# Genome-wide analyses identify 25 infertility loci and relationships with reproductive traits across the allele frequency spectrum

Genome-wide association studies (GWASs) may help inform the etiology of infertility. Here, we perform GWAS meta-analyses across seven cohorts in up to 42,629 cases and 740,619 controls and identify 25 genetic risk loci for male and female infertility. We additionally identify up to 269 genetic loci associated with follicle-stimulating hormone, luteinizing hormone, estradiol and testosterone through sex-specific GWAS meta-analyses ($n$ = 6,095–246,862). Exome sequencing analyses reveal that women carrying testosterone-lowering rare variants in some genes are at risk of infertility. However, we find no local or genome-wide genetic correlation between female infertility and reproductive hormones. While infertility is genetically correlated with endometriosis and polycystic ovary syndrome, we find limited genetic overlap between infertility and obesity. Finally, we show that the evolutionary persistence of infertility-risk alleles may be explained by directional selection. Taken together, we provide a comprehensive view of the genetic determinants of infertility across multiple diagnostic criteria.

Infertility, defined as the inability to achieve pregnancy within 12 months of regular unprotected sexual intercourse, affects one in six couples across the globe[1]. A range of demographic, environmental and genetic factors may drive infertility, including the age-related decline of sperm and oocyte quality and quantity, infectious diseases and rare Mendelian disorders such as cystic fibrosis. However, the exact cause remains undetermined in up to 28% of couples and 40% of women with infertility[2]. Given that current treatments such as in vitro fertilization pose physical, emotional and financial burdens on couples and healthcare systems, a richer understanding of the biology and pathophysiology of infertility is urgently necessary.

Heritable women's reproductive health diseases such as endometriosis[3] and polycystic ovary syndrome (PCOS)[4] are thought to be responsible for a considerable proportion of female infertility, with PCOS in particular accounting for up to 80% of cases of anovulatory infertility[4]. It is hypothesized that sex-hormone dysregulation[5,6] and obesity[7], which often accompany reproductive diseases, may be involved in the etiology of infertility. Yet little is known about the

genetic basis of reproductive hormones and infertility, which are not well phenotyped in men or women in large studies[8,9]. Moreover, negative selection against infertility naturally limits the frequency of risk alleles in the population. Genome-wide association studies (GWASs) have thus typically queried proxy measures of fertility such as childlessness[10,11], which may partly arise from socioeconomic and behavioral factors.

We aggregated data from a range of sources, including primary care and hospital electronic health records and self-report, across seven cohorts with over 1.5 million participants, to perform GWAS meta-analyses for male infertility and five categories of female infertility. In addition, we report results from the largest sex-specific GWASs so far for five reproductive hormones. By aggregating these data with complementary rare-variant genetic association testing, we catalog the common and rare genetic contributions to infertility and reproductive hormone levels, quantify the extent of shared genetic architecture between these traits and prioritize genes for further functional investigation of the hormonal and non-hormonal drivers of infertility.

✉e-mail: samvida.venkatesh@gmail.com; cecilia.lindgren@bdi.ox.ac.uk

## Results

### Genome-wide meta-analyses identify new loci for infertility

We identified female infertility of all causes (F-ALL), anatomical causes (F-ANAT), anovulation (F-ANOV), unknown causes (that is, idiopathic infertility as defined by exclusion of known causes of infertility (anatomical or anovulatory causes, PCOS, endometriosis or uterine leiomyomas)) (F-EXCL) or idiopathic infertility defined by inclusion of diagnostic codes for idiopathic infertility (F-INCL), as well as male infertility of all causes (M-ALL) in seven cohorts, primarily of European ancestry (EUR) (Fig. 1 and Supplementary Tables 1 and 2). The case–control ratio of all-cause female infertility ranged from 0.9% in the deCODE Genetics dataset to 11.7% in FinnGen, whereas the case–control ratio of male infertility was between 0.3% (UK Biobank (UKBB)) and 8.2% (Danish Biobank) (Fig. 1 and Supplementary Table 2). Anatomical female infertility was the least common cause of infertility in three of six cohorts (prevalence in UKBB of 0.01%, FinnGen of 0.8% and Estonian Biobank (EstBB) of 2.0%). Owing to varying sample ascertainment, the case–control ratio does not necessarily reflect the population prevalence of infertility.

### Novel genetic loci for infertility

We performed GWAS meta-analyses, testing up to 33 million genetic variants for associations with each of the above categories of infertility, in up to 42,629 cases and 740,619 controls in women, and 10,886 cases and 995,982 controls in men (Fig. 1 and Supplementary Table 2). We identified 22 unique genome-wide significant ($P < 5 \times 10^{-8}$) loci associated with at least one category of female infertility and three loci for male infertility (minor allele frequency (MAF) range 0.06–46%) (Fig. 2, Table 1 and Supplementary Fig. 1). Fourteen loci (63.6%) for female infertility reached nominal significance ($P < 2.27 \times 10^{-3}$, Bonferroni correction for 22 independent loci tested) in at least one other infertility category (Supplementary Note and Supplementary Fig. 19). There was no evidence for heterogeneity in lead variant effects across cohorts (Supplementary Note and Supplementary Table 3).

Among the variants associated with multiple subtypes of female infertility is rs9643050 (MAF of 6.01%), an intronic variant in *PKHD1L1* (F-ALL, odds ratio (OR) (95% confidence interval (CI)) 1.13 (1.09–1.16); F-EXCL, OR 1.13 (1.09–1.17); F-INCL, OR 1.18 (1.11–1.25)). This variant is 76 kb upstream of *EBAG9*, an estrogen-responsive gene previously reported to have a recessive association with female infertility[12,13] and thought to suppress maternal immune response during pregnancy[14,15]. We also identified an intronic variant in *WNT4*, rs61768001 (MAF of 16.5%), associated with three categories of female infertility (F-ALL, OR 1.10 (1.08–1.12); F-EXCL, OR 1.08 (1.06–1.11); F-INCL, OR 1.15 (1.11–1.19)). *WNT4* is highly pleiotropic for female reproductive traits, as it is reported to associate with gestational length[16], uterine fibroids[17,18], endometriosis[19,20], female genital prolapse[21] and bilateral oophorectomy[21]. Such pleiotropy reflects the role of *WNT4* as a key regulator of female reproductive organ development during embryogenesis[22].

The nearest gene to the idiopathic infertility-associated variant rs111597692 (MAF of 3.23%; F-EXCL, OR 1.16 (1.10–1.22)) is *TRHR*, which encodes the thyrotropin-releasing hormone receptor. Mice with *Trhr* knockout display a phenotype similar to primary ovarian insufficiency[23]. The F-ANOV-associated variant rs72827480 (MAF of 40.1%, OR 1.10 (1.07–1.14)) colocalizes with a testis expression quantitative trait locus (eQTL) for *INHBB* in the GTEx Project (posterior probability (PP) of shared causal variant of 91.6%; Supplementary Table 4). *INHBB* encodes the beta subunit of inhibin B, which regulates hypothalamic, pituitary and gonadal hormone secretion[24], and ovarian follicle and oocyte development[25]. rs111749498 (MAF of 2.73%, associated with F-ALL, OR 2.29 (1.72–3.04)) is near *SLC47A2*, which encodes a multidrug efflux pump that mediates excretion of the drug metformin, commonly used to manage infertility in women with PCOS[26]. Variants associated with all-cause female infertility are in genes enriched for expression in ovarian stromal cells (partitioned heritability $P = 2.52 \times 10^{-3}$; Supplementary Note).

The male infertility-associated variant rs75957543 (MAF of 1.25%, OR 1.67 (1.39–2.01)) is near *UMODL1*, which encodes the olfactorin protein, expressed along the migratory route of gonadotropin-releasing hormone neurons. Impairment of gonadotropin-releasing hormone migration is a feature of Kallmann's syndrome, the most common genetic cause of hypogonadotropic infertility[27]. While mutations in *UMODL1* have been shown to impact ovarian follicle development, granulosa cell apoptosis and female fertility in model organisms[28,29], its role in male infertility remains unclear. Finally, an intronic variant in *ENO4*, which is expressed in the testis and may play a role in sperm motility[30], is associated with male infertility (rs139862664, MAF of 0.72%, OR 2.58 (1.84–3.60)). Male mice with *Eno4* knockout display infertility, abnormal sperm morphology and physiology and decreased testis weight, among other altered male reproductive tract phenotypes[31].

### Relationships with other female reproductive conditions

Genome wide, we observed positive genetic correlations (Fig. 3a) between endometriosis and F-ALL ($r_g$ (s.e.m.) = 0.585 (0.0785), $P = 8.98 \times 10^{-14}$) and F-INCL ($r_g = 0.710$ (0.115), $P = 5.94 \times 10^{-10}$). We also observed positive correlation between F-ANOV and PCOS, the most common cause of anovulatory infertility ($r_g = 0.403$ (0.131), $P = 2.20 \times 10^{-3}$). We tested for local bivariate genetic correlations between infertility and PCOS, endometriosis, heavy menstrual bleeding and uterine fibroids at 2,495 blocks across the genome, chosen to be approximately 1 Mb in length each, while minimizing linkage disequilibrium (LD) between blocks. Consistent with the genome-wide $r_g$, we found positive local $r_g$ between female infertility and reproductive disorders at 11 regions ($P < 1.91 \times 10^{-5}$, Bonferroni adjustment for 2,618 local bivariate tests performed at regions with significant heritability of both traits in each pair tested; Fig. 4a and Supplementary Table 22). At 5/11 blocks, infertility was correlated with more than one reproductive condition, none of which had individual effects after conditioning upon the other associated reproductive disorders in the region (all $P > 0.05$; Supplementary Table 22).

Furthermore, we used MiXeR[32] to assess bivariate polygenic overlap, regardless of genome-wide genetic correlation, between infertility and reproductive conditions. We found that approximately 50% of causal single-nucleotide polymorphisms (SNPs) involved in endometriosis, and about 25% of causal SNPs involved in uterine fibroids were shared with the assessed infertility phenotypes, with varying degrees of genetic correlation in the shared component (Fig. 4b, Supplementary Table 24 and Supplementary Note). We noted that while there was substantial correlation in the shared component of F-ANOV and PCOS (rho (s.e.m.) of 0.878 (0.242)), only 97 (10.9%) of the 888 causal variants involved were shared; the majority (88.2%) of variants were unique to F-ANOV and only 8 variants (<1%) were unique to PCOS, suggesting that a small proportion of causal variants drive the genetic correlation between these traits (Fig. 4b and Supplementary Table 24).

We observed genome-wide negative correlation between F-ANOV and spontaneous dizygotic twinning, a heritable metric of female fecundity that captures the propensity for multiple ovulation[33] ($r_g = -0.740$ (0.182), $P = 4.93 \times 10^{-5}$). We also found substantial negative correlation in the shared polygenic component of these traits (rho (s.e.m.) = -0.920 (0.129)), with 32% (295) shared SNPs of the 912 total causal SNPs involved (Fig. 4b, Supplementary Table 24 and Supplementary Note).

Two loci associated with both endometriosis and female infertility (*WNT4* and *ESR1*) may share the same putative causal variant (PP >93.6%; Supplementary Table 5). Variants in both these genes have previously been associated with endometriosis-related infertility[34,35]. *GREB1* and *SYNE1* also contain overlapping signals for infertility and endometriosis, but there is strong evidence against shared causal variants (PP >75%; Supplementary Table 5). Finally, three of eight loci for

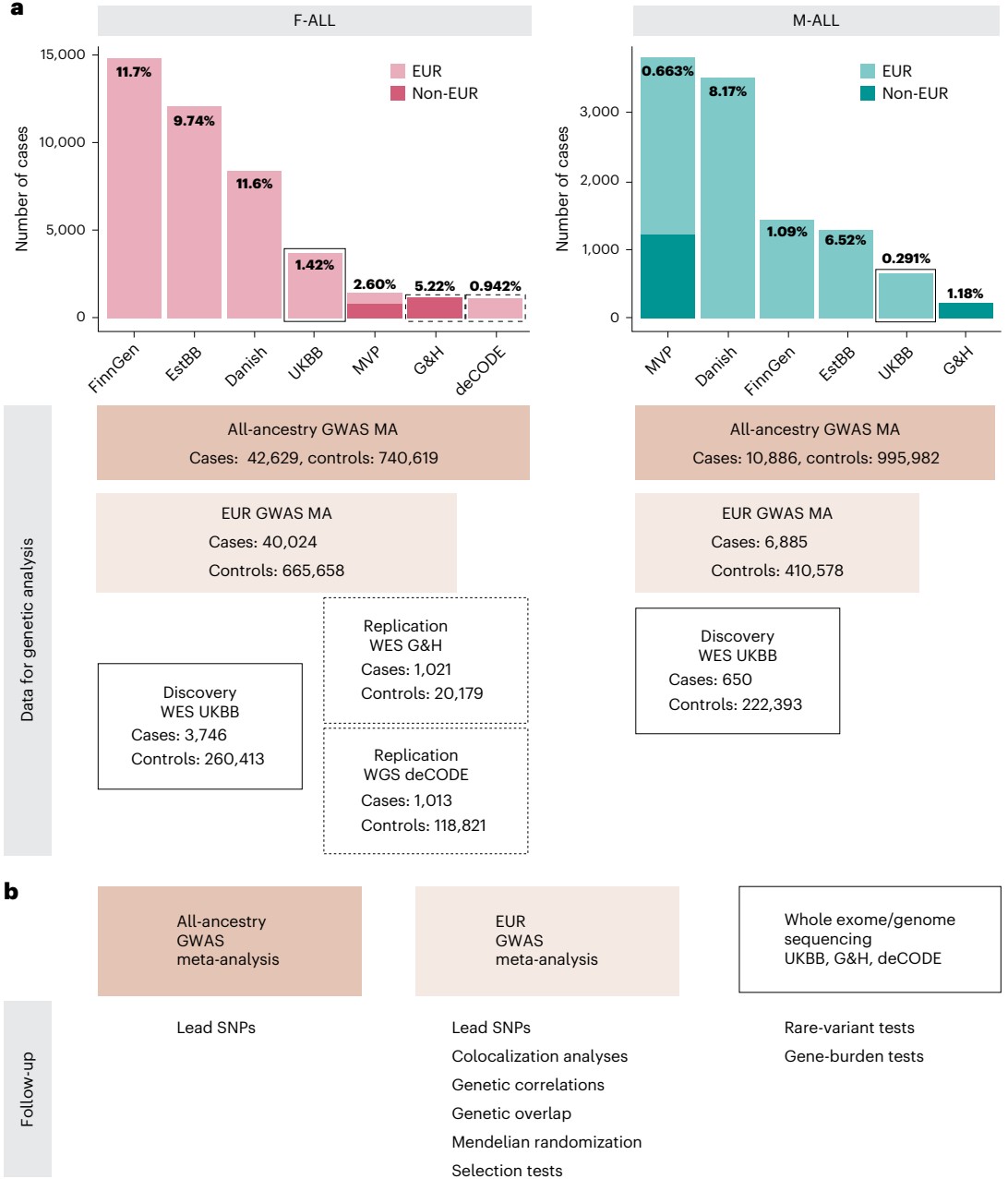

**Fig. 1 | Overview of study cohorts and analyses for infertility genetic association studies. a**, The case numbers in each cohort contributing to GWAS meta-analyses (MA) for female (left) and male (right) infertility. The prevalence of all-cause infertility in each cohort (%) is noted on the bar plots. Danish, Danish Blood Donor Study/Copenhagen Hospital Biobank. Total case and control counts for each type of genetic analysis: all-ancestry GWAS meta-analysis, EUR-only GWAS meta-analysis and WES analyses (discovery, UKBB and replication, G&H and deCODE) are displayed. Male infertility in deCODE, with <100 cases, was excluded from GWAS meta-analysis. Note the different *y*-axis scales in each subplot. **b**, Downstream analyses performed for each type of genetic analysis: lead variants were identified via distance-based pruning for all-ancestry and EUR-only GWAS meta-analyses; colocalization, genetic correlations (genome wide and local), genetic overlap and selection analyses were only performed for EUR meta-analyses due to the need for ancestry-matched LD information; rare-variant and gene-burden discovery tests were performed with WES data for the UKBB EUR-ancestry subset and replicated in individuals with WES data in G&H and whole-genome sequencing (WGS) data in deCODE.

anovulatory infertility (*INHBB*, *TTC28* and *CHEK2*) may share a causal variant with PCOS (PP >89.2%; Supplementary Table 5).

**Evolutionary persistence of infertility-associated variants**
The genome-wide SNP heritability estimates (on the liability scale, accounting for disease prevalence) for all categories of infertility were <10% (lowest for M-ALL at 1.12% (s.e.m. 0.93) and highest for F-ANOV at 9.54% (s.e.m. 2.16); Supplementary Table 6). This is lower than heritability estimates of two-thirds of all heritable binary phenotypes in the UKBB, with population prevalence similar to that of infertility (64 phenotypes with $Z > 4$ and prevalence <5%)[36]. We hypothesized that infertility risk-increasing alleles are subject to negative selection[37], so we tested whether there was evidence for (1) variants associated with infertility in loci under historical or recent directional selection[38] or (2) recent directional selection (over the past 2,000–3,000 years) measured by singleton density scores (SDSs)[39] and balancing selection measured by standardized BetaScan2 scores[40] at infertility loci (Supplementary Note).

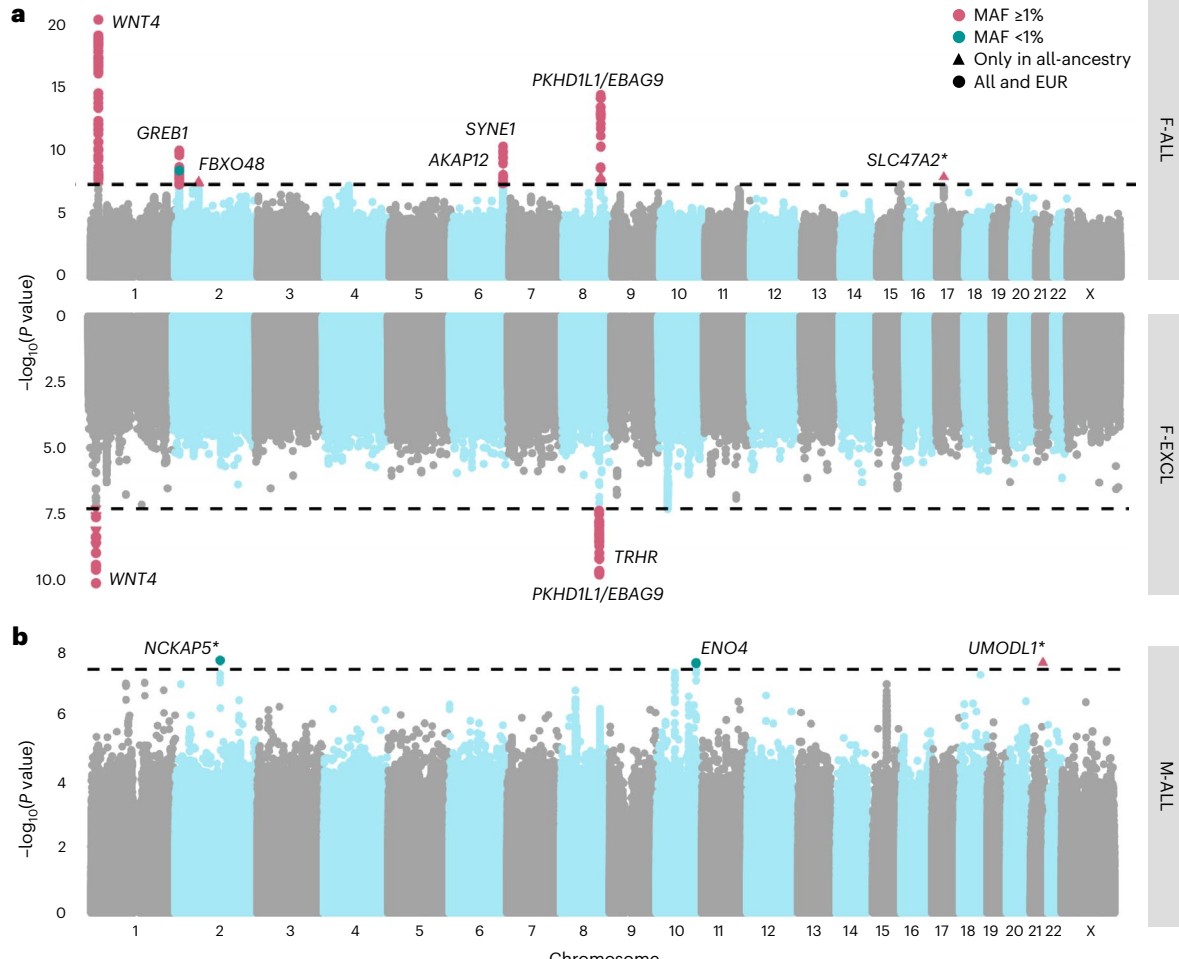

**Fig. 2 | Miami and Manhattan plots for selected infertility meta-analyses.**
**a**, Genetic variants associated with F-ALL (top) and idiopathic infertility (unknown causes) defined by exclusion of known causes such as anatomical or anovulatory causes, PCOS, endometriosis and uterine leiomyomas (bottom). **b**, Genetic variants associated with M-ALL. Each point depicts a single SNP. The triangles represent SNPs that only reach genome-wide significance in all-ancestry GWAS meta-analyses. SNPs are annotated with the mapped gene. *The lead variant is reported in only one cohort. Summary statistics from whole-genome regression analyses were meta-analyzed using fixed-effect inverse-variance weighting in the METAL software to produce the displayed $P$ values. The dashed line represents the multiple testing-corrected $P$ value threshold of $P < 5 \times 10^{-8}$, accounting for ~1 million independent variants in the genome.

While we found no genome-wide signature of directional selection against infertility (Supplementary Note), we observed extreme SDSs (in the highest 99.75th percentile of SNPs within 10 kb of a GWAS catalog variant) at the *EBAG9* locus associated with female infertility, indicating recent positive selection (Fig. 5 and Supplementary Table 7).

**Genetic determinants of reproductive hormone levels**
**Identification of novel reproductive hormone loci.** As hormone dysregulation is central to many infertility diagnoses[5,6], we conducted sex-specific GWAS meta-analyses of five reproductive hormones—follicle-stimulating hormone (FSH) ($n_{female} = 57,890$, $n_{male} = 6,095$), luteinizing hormone (LH) ($n_{female} = 47,986$, $n_{male} = 6,769$), estradiol ($n_{female} = 97,887$, $n_{male} = 39,165$), progesterone ($n_{female} = 18,368$) and total testosterone ($n_{female} = 246,862$, $n_{male} = 243,951$)—collected at assessment center visits or identified through electronic health records, in six cohorts and publicly available summary statistics (Supplementary Table 9). We identified genome-wide significant loci associated with FSH (9 novel/2 previously known in females and 0/1 in males), LH (4/2 in females and 1/0 in males), estradiol (1/1 in females and 3/4 in males) and testosterone (39/118 in females and 67/206 in males), but found no genetic variants associated with progesterone (Supplementary Figs. 3,

4 and 20). Several of the reported signals we replicated are near genes encoding the hormone-specific subunits themselves, such as *FSHB* for FSH and *LHB* for LH, or enzymes for steroid-hormone metabolism, such as *CYP3A7* for estradiol and *HSD17B13* for testosterone (Supplementary Note).

Among the novel variants for testosterone in men were those near *SPOCK1* (rs1073917: MAF of 30.7%, $\beta$ (s.e.m.) = 0.0160 (0.0029), $P = 4.69 \times 10^{-8}$), which is a target for the androgen receptor[41], and *NR4A3* (rs10988865: MAF of 27.4%, $\beta$ = 0.0161 (0.0029), $P = 4.33 \times 10^{-8}$), which coordinates the cellular response to corticotropin hormone- and thyrotropin hormone-releasing stimuli[42] (Supplementary Table 10). Novel reproductive hormone variants associated with testosterone in women include those near *LAMTOR4* (rs17250196: MAF of 5.13%, $\beta$ = −0.131 (0.0067), $P = 4.02 \times 10^{-86}$), associated with hyperthyroidism[23] and age at menarche and menopause[43], and obesity-associated *CCDC146* (rs138240474: MAF of 0.63%, $\beta$ = −0.116 (0.0207), $P = 2.03 \times 10^{-8}$)[44], which is expressed in the fallopian tubes and endometrium[45].

Clinical measurements of FSH and LH may be used to diagnose premature menopause[46], but our hormone GWASs based on these measurements were robust to this potential ascertainment bias (Supplementary Note). They were also robust to the inclusion of summary

**Table 1 | Lead variants associated with infertility on GWAS meta-analyses**

| rsID | chr:pos:A1:A2 (hg38) | Mapped gene | All ancestries | | | EUR only | | |
|---|---|---|---|---|---|---|---|---|
| | | | Average MAF | OR (95% CI) | *P* value | Average MAF | OR (95% CI) | *P* value |
| **Female infertility of all causes (F-ALL)** | | | | | | | | |
| rs61768001 | chr1:22139327:C:T | *WNT4* | 0.165 | 1.10 (1.08–1.12) | $4.27×10^{-21}$ | 0.163 | 1.10 (1.08–1.12) | $1.24×10^{-19}$ |
| rs10929759 | chr2:11581886:G:C | *GREB1* | 0.456 | 0.952 (0.938–0.966) | $1.03×10^{-10}$ | 0.456 | 0.951 (0.937–0.966) | $8.03×10^{-11}$ |
| rs75045132 | chr2:68456386:C:G | *FBXO48* | 0.0331 | 0.888 (0.851–0.926) | $3.20×10^{-8}$ | 0.0328 | 0.889 (0.852–0.929) | $1.20×10^{-7}$ |
| rs6938404 | chr6:151222906:T:C | *AKAP12* | 0.452 | 0.958 (0.944–0.973) | $4.01×10^{-8}$ | 0.453 | 0.958 (0.943–0.973) | $3.88×10^{-8}$ |
| rs17803970 | chr6:152232583:T:A | *SYNE1* | 0.0833 | 0.914 (0.889–0.939) | $5.06×10^{-11}$ | 0.0836 | 0.909 (0.885–0.934) | $7.50×10^{-11}$ |
| rs9643050 | chr8:109458129:C:T | *PKHD1L1* | 0.0601 | 1.13 (1.10–1.16) | $3.98×10^{-15}$ | 0.0597 | 1.13 (1.09–1.16) | $7.43×10^{-14}$ |
| rs111749498* | chr17:19714694:T:C | *SLC47A2* | 0.0273 | 2.29 (1.72–3.04) | $1.29×10^{-8}$ | – | – | – |
| **Anatomical female infertility (F-ANAT)** | | | | | | | | |
| rs340879 | chr1:213983171:T:C | *PROX1* | 0.418 | 0.906 (0.874–0.939) | $4.95×10^{-8}$ | 0.418 | 0.902 (0.869–0.936) | $5.06×10^{-8}$ |
| **Anovulatory female infertility (F-ANOV)** | | | | | | | | |
| rs72665317 | chr1:22040580:G:T | *CDC42* | 0.190 | 1.14 (1.10–1.19) | $7.76×10^{-10}$ | 0.180 | 1.13 (1.08–1.18) | $1.45×10^{-7}$ |
| rs72827480 | chr2:120388925:C:T | *INHBB* | 0.401 | 1.10 (1.07–1.14) | $4.20×10^{-8}$ | 0.401 | 1.10 (1.07–1.14) | $4.20×10^{-8}$ |
| rs1852684 | chr2:145068818:T:G | *ZEB2* | 0.367 | 1.12 (1.08–1.16) | $9.25×10^{-10}$ | 0.350 | 1.12 (1.08–1.17) | $3.44×10^{-10}$ |
| rs552953683 | chr8:102898586:C:T | *AZIN1* | 0.0024 | 2.93 (2.01–4.27) | $2.54×10^{-8}$ | 0.0024 | 2.93 (2.01–4.27) | $2.54×10^{-8}$ |
| rs9696009 | chr9:123856954:A:G | *DENND1A* | 0.0777 | 1.21 (1.14–1.29) | $6.87×10^{-10}$ | 0.0695 | 1.24 (1.16–1.32) | $2.40×10^{-10}$ |
| rs9902027 | chr17:7537667:C:T | *TNFSF12* | 0.255 | 1.12 (1.07–1.16) | $4.06×10^{-8}$ | 0.255 | 1.12 (1.07–1.16) | $4.06×10^{-8}$ |
| rs143459581 | chr22:28068862:T:C | *TTC28* | 0.0419 | 1.30 (1.19–1.43) | $1.21×10^{-8}$ | 0.0419 | 1.30 (1.19–1.43) | $1.21×10^{-8}$ |
| rs17879961 | chr22:28725099:G:A | *CHEK2* | 0.0389 | 1.35 (1.23–1.48) | $1.55×10^{-10}$ | 0.0389 | 1.35 (1.23–1.48) | $1.55×10^{-10}$ |
| **Idiopathic female infertility, exclusion definition (F-EXCL)** | | | | | | | | |
| rs61768001 | chr1:22139327:C:T | *WNT4* | 0.165 | 1.08 (1.06–1.11) | $7.49×10^{-11}$ | 0.162 | 1.08 (1.05–1.10) | $2.48×10^{-9}$ |
| rs111597692 | chr8:109039973:T:C | *TRHR* | 0.0323 | 1.16 (1.10–1.22) | $1.51×10^{-8}$ | 0.0323 | 1.16 (1.1–1.22) | $1.51×10^{-8}$ |
| rs17378154 | chr8:109568721:A:G | *PKHD1L1* | 0.0590 | 1.13 (1.09–1.17) | $1.64×10^{-10}$ | 0.0593 | 1.13 (1.09–1.17) | $3.36×10^{-10}$ |
| **Idiopathic female infertility, inclusion definition (F-INCL)** | | | | | | | | |
| rs61768001 | chr1:22139327:C:T | *WNT4* | 0.170 | 1.15 (1.11–1.19) | $6.87×10^{-14}$ | 0.165 | 1.15 (1.10–1.19) | $8.96×10^{-13}$ |
| rs11692588 | chr2:11544358:A:G | *GREB1* | 0.358 | 0.919 (0.892–0.947) | $2.98×10^{-8}$ | 0.358 | 0.919 (0.892–0.947) | $2.98×10^{-8}$ |
| rs190290095 | chr4:39786858:A:G | *UBE2K* | 0.0022 | 0.227 (0.137–0.375) | $7.60×10^{-9}$ | 0.0022 | 0.227 (0.137–0.375) | $7.60×10^{-9}$ |
| rs851982 | chr6:151703850:C:T | *ESR1* | 0.428 | 1.08 (1.06–1.12) | $7.60×10^{-9}$ | 0.437 | 1.08 (1.05–1.12) | $2.86×10^{-8}$ |
| rs17378154 | chr8:109568721:A:G | *PKHD1L1* | 0.0565 | 1.18 (1.11–1.25) | $2.47×10^{-8}$ | 0.0569 | 1.18 (1.11–1.25) | $4.97×10^{-8}$ |
| rs74156208 | chr10:61509370:A:G | *TMEM26* | 0.184 | 1.10 (1.06–1.14) | $4.96×10^{-8}$ | 0.187 | 1.10 (1.07–1.15) | $5.44×10^{-8}$ |
| rs192462512* | chrX:39653668:C:T | *BCOR* | 0.0015 | 0.227 (0.142–0.363) | $5.26×10^{-10}$ | 0.0015 | 0.227 (0.142–0.363) | $5.26×10^{-10}$ |
| **Male infertility of all causes (M-ALL)** | | | | | | | | |
| rs1228269928* | chr2:132923776:T:A | *NCKAP5* | 0.0006 | 10.0 (4.45–22.7) | $2.72×10^{-8}$ | 0.0006 | 10.0 (4.45–22.7) | $2.72×10^{-8}$ |
| rs139862664 | chr10:116879589:G:C | *ENO4* | 0.0072 | 2.58 (1.84–3.60) | $3.29×10^{-8}$ | 0.0072 | 2.58 (1.84–3.60) | $3.29×10^{-8}$ |
| rs75957543* | chr21:42081234:G:C | *UMODL1* | 0.0125 | 1.67 (1.39–2.01) | $3.19×10^{-8}$ | – | – | – |

Summary statistics from whole-genome regression analyses were meta-analyzed using fixed-effect inverse-variance weighting in the METAL software. Lead variants were identified by distance-based pruning within windows of 1Mb at all loci with at least one variant with $P<5×10^{-8}$ (multiple testing correction for ~1 million independent variants in the genome). A1 is the minor (effect) allele. Exonic or intronic variants in coding genes are mapped to their genes; intergenic variants are mapped to the nearest coding gene (by TSS). *Lead variant is reported in only one cohort. Blank cells indicate that the variant was not present in the EUR-only meta-analysis.

statistics from publicly available datasets and there was no evidence for heterogeneity in variant effects across cohorts (Supplementary Note).

**Relationships with other heritable phenotypes**

We observed no genome-wide genetic correlations between any category of female infertility and (1) any reproductive hormone in this study, (2) thyroid stimulating hormone (TSH) or (3) anti-Mullerian hormone (AMH), the latter two based on publicly available summary statistics[47,48] (all $P > 0.05$, except the correlation between AMH and F-ANOV, $r_g$ (s.e.m.) = 0.748 (0.301), $P = 0.0131$; Fig. 3b). Consistent with

the genome-wide results, we also found no evidence for local genetic correlations between any category of infertility and the above hormones (all $P > 1.91×10^{-5}$; Fig. 4a, Supplementary Table 22 and Supplementary Note). The limited genetic correlation between infertility and reproductive hormones was mirrored in polygenic overlap analyses. The highest proportion of shared SNPs between these traits was 14.5% between F-ANOV and testosterone (209/1,444 variants shared, rho (s.e.m.) of 0.549 (0.252) in the shared polygenic component), followed by 14.0% between F-ANOV and AMH (123/881, rho (s.e.m.) of 0.993 (0.0301); Fig. 4b and Supplementary Table 24).

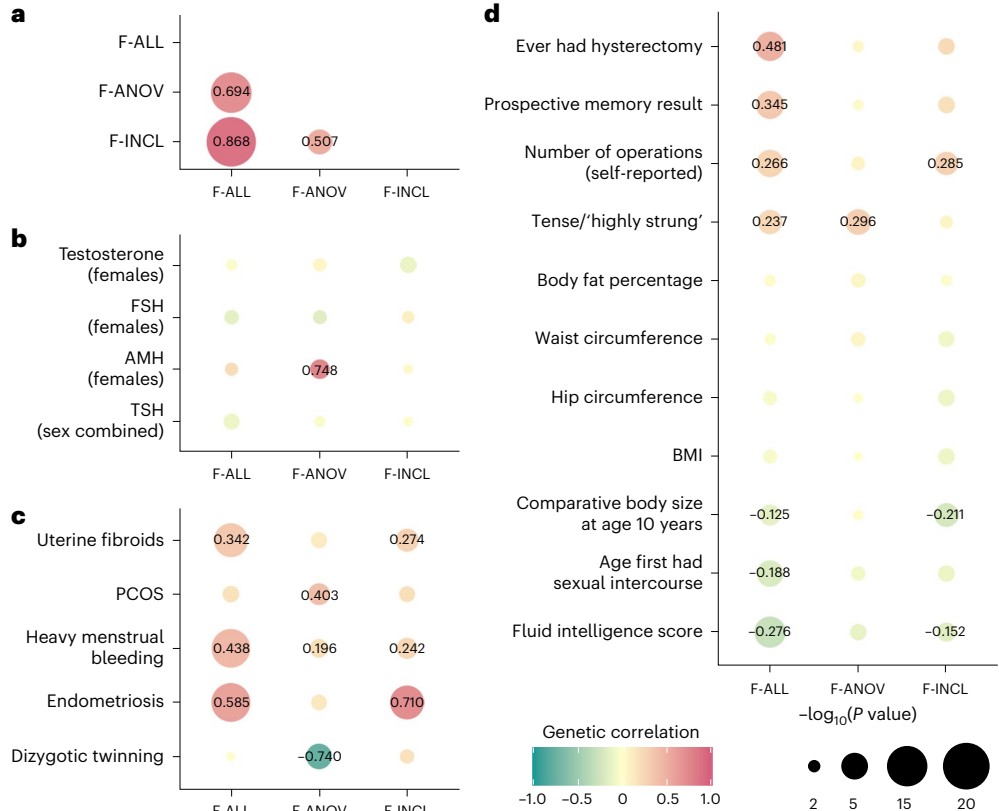

**Fig. 3 | Genetic correlations between female infertility and other phenotypes.** SNP-based genetic correlations ($r_g$) between significantly heritable phenotypes ($Z > 4$) were estimated using LD-score regression, performed using the LDSC software on a subset of 1 million HapMap3 SNPs. The points are colored by $r_g$ estimate, scaled by significance ($-\log_{10}(P)$), and labeled with the associated $r_g$ estimate if nominally significant without correction for multiple testing ($P < 0.05$). **a**, Genetic correlations among three definitions of female infertility (F-ALL, F-ANOV and F-INCL). **b**, Genetic correlations between female infertility traits and

reproductive hormones testosterone, FSH and AMH (publicly available summary statistics) in female-specific analyses and TSH (publicly available summary statistics) from sex-combined analysis. **c**, Genetic correlations between female infertility traits and female reproductive conditions, with summary statistics generated from the largest available EUR-ancestry studies for each trait (Methods). **d**, Genetic correlations between female infertility traits and selected heritable phenotypes ($Z > 4$) in the UKBB, as generated by the Neale laboratory. Correlations with all heritable phenotypes can be found in Supplementary Table 12.

Mendelian randomization (MR) analyses indicated a genetically causal protective effect of FSH on risk of F-ALL (OR (95% CI) 0.776 (0.678–0.888), $P = 2.15 \times 10^{-4}$) and F-EXCL (0.716 (0.604–0.850), $P = 1.26 \times 10^{-4}$) (Supplementary Table 11). We found evidence for shared variants between hormones and infertility at the *FSHB* locus associated with FSH, LH and testosterone (PP >84.8% for colocalization with F-ANOV), and the *ARL14EP* locus associated with LH (PP 89.3% for colocalization with F-ANOV) (Supplementary Table 12). There was no evidence for colocalization at any of the >300 other genome-wide significant loci associated with infertility or reproductive hormones in our study (Supplementary Table 12).

Across 702 heritable phenotypes in the UKBB, we found 15 traits to be genetically correlated with female infertility, which we broadly group into: female reproductive conditions (such as having had a hysterectomy, $r_g$ (s.e.m.) = 0.481 (0.0963)), general illness (such as number of operations, $r_g$ = 0.266 (0.0588)), and cognitive test results (overall prospective memory test $r_g$ = 0.345 (0.0736) and overall fluid intelligence $r_g$ = −0.276 (0.0502)) (Fig. 3d and Supplementary Table 13). We found that 24 obesity-related traits, including body mass index (BMI), waist-to-hip ratio (WHR) and body fat percentage, were correlated with testosterone and FSH, but not with any category of female infertility (all $P > 0.05$; Fig. 3d, Supplementary Fig. 7 and Supplementary Table 13).

We found no evidence for local genetic correlations between any category of infertility and five obesity-related traits at 2,495 regions across the genome at a Bonferroni-adjusted significance threshold

(all $P > 1.91 \times 10^{-5}$; Fig. 4a, Supplementary Table 22 and Supplementary Note). Polygenic analyses also revealed only limited overlap between infertility and obesity: fewer than 10% of causal SNPs involved were shared between infertility and any of the five obesity-related traits assessed (Fig. 4b, Supplementary Table 24 and Supplementary Note). The low overlap may reflect the polygenicity of obesity (between 4,050 and 11,000 causal variants), of which the majority (between 73.2% and 93.0%) are not involved in infertility (Supplementary Tables 23 and 24). Despite limited overlap, there was substantial negative correlation in the shared genetic components between F-INCL and comparative body size at age 10 years (451 shared SNPs of 4,385 involved, rho (s.e.m.) of −0.874 (0.143)) and adult BMI (393/11,185, rho (s.e.m.) of −0.640 (0.262)) (Fig. 4b and Supplementary Table 24).

Finally, MR analyses using genetic instruments for BMI, WHR and WHR adjusted for BMI (WHRadjBMI) indicated evidence for bidirectional causal relationships between infertility and abdominal obesity, independent of overall obesity. While genetically predicted WHRadjBMI was a risk factor for F-ALL (OR (95% CI) 1.10 (1.05–1.16), $P = 1.71 \times 10^{-4}$) and F-ANOV (OR 1.29 (1.16–1.45), $P = 4.66 \times 10^{-6}$), the latter was itself inferred to be causal for increased WHRadjBMI ($\beta$ (s.e.m.) = 0.0547 (0.0133), $P = 3.74 \times 10^{-5}$) (Supplementary Table 11).

**Rare-variant contribution to infertility and hormone levels**

We analyzed the 450k UKBB exome-sequencing dataset to characterize the association between rare coding variation (MAF <1%) and binary traits with >100 cases (F-ALL (3,746 cases, 260,413 controls),

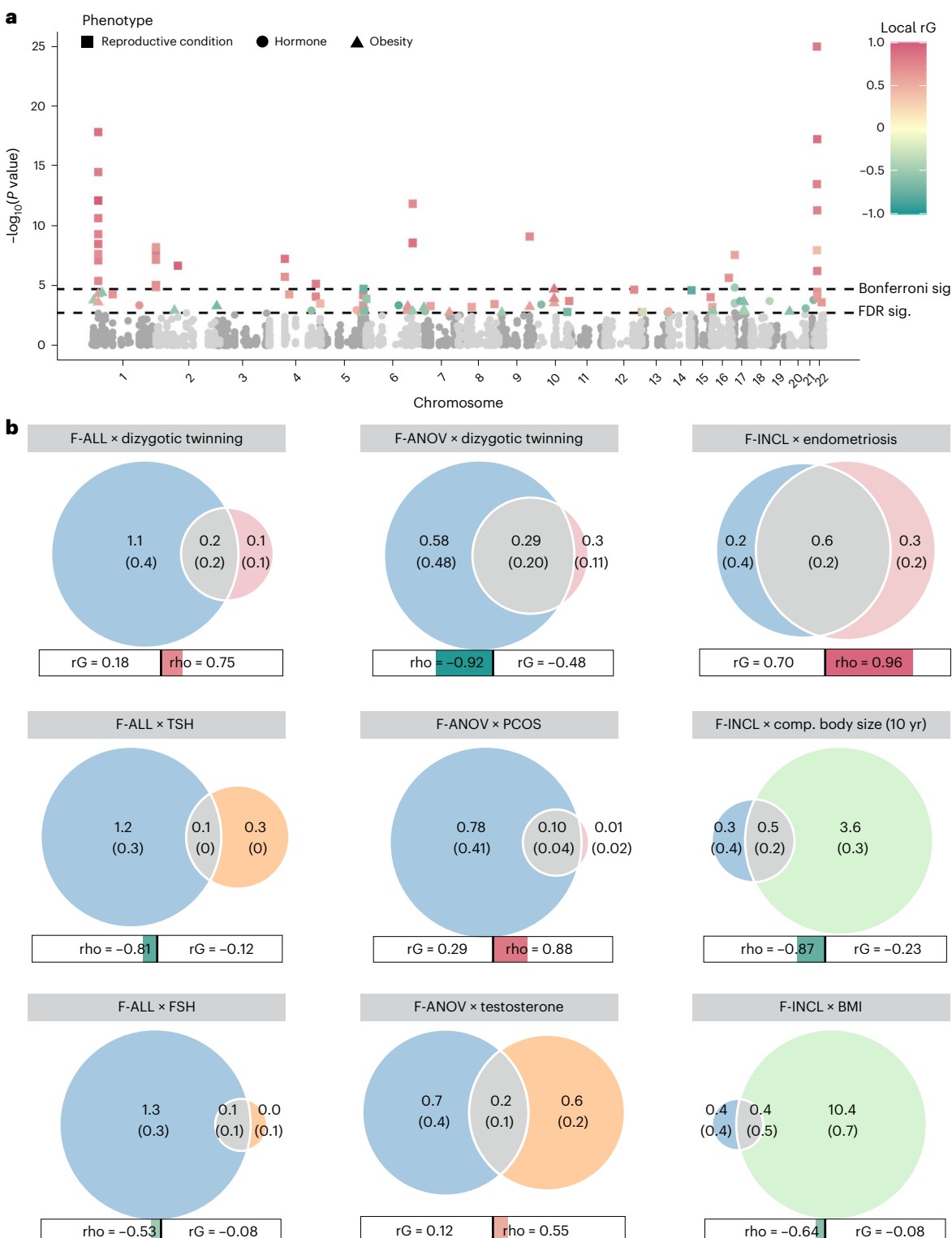

**Fig. 4 | Local genetic correlations and polygenic overlap between female infertility and other phenotypes. a**, Local genetic correlations, estimated using LAVA, at 2,495 blocks across the genome. Each point represents a local bivariate genetic correlation between an infertility trait (F-ALL, F-ANOV or F-INCL) and reproductive hormone, reproductive condition or obesity-related trait. The dashed lines indicate significance (sig.) thresholds. The dashed line represents FDR-adjusted or Bonferroni-adjusted *P* values of 0.05. **b**, MiXeR estimates of polygenic overlap. The Venn diagrams indicate the estimated number (s.e.m.) of causal variants (in thousands) that explain 90% SNP heritability per component. The circle size reflects the degree of polygenicity. The bars outline the genome-wide genetic correlation (rG) and correlation in the shared polygenic component (rho). The colored portion of the bar is sized by the proportion of causal variants in the shared polygenic component as compared with all causal variants involved and colored by rho. Comp., comparative.

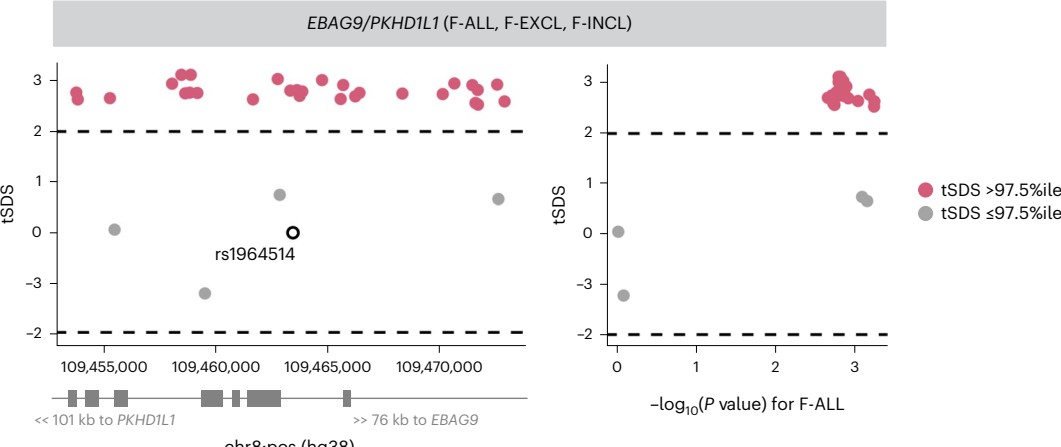

**Fig. 5 | Directional selection scores at infertility-associated *EBAG9* locus.**
Recent directional selection, as measured by trait-aligned SDSs (tSDSs) at the *EBAG9* locus. The window of ±10 kb around the lead variant associated with F-ALL is displayed, along with the location of nearest gene TSSs. The tSDSs are aligned to the infertility risk-increasing allele, wherein a positive tSDS indicates positive selection for infertility risk-increasing allele at the locus. The dashed lines indicate 2.5th percentile (%ile) and 97.5th %ile of SDSs. Left: a locus plot depicting genomic position on the *x* axis and tSDS on the *y* axis. The lead variant rs1964514 (open circle) is not present in the tSDS dataset and thus is assigned a score of 0. Right: a scatter plot depicting relationship between $-\log_{10}$ of the GWAS *P* value for the variant association with F-ALL on the *x* axis and tSDS on the *y* axis.

F-EXCL (3,012 cases, 261,147 controls) and M-ALL (650 cases, 222,393 controls)), and quantitative traits with >10,000 participants (FSH-F ($n = 20,800$), LH-F ($n = 16,391$), estradiol-F ($n = 54,609$) and testosterone ($n_{\text{female}} = 197,038$, $n_{\text{male}} = 197,340$)) (Fig. 1). Gene-burden analyses implicate the *PLEKHG4* gene, which is highly expressed in the testis and ovary, for F-EXCL (burden test OR (95% CI) 1.04 (1.02–1.06) when aggregated across all variant annotations with MAF <1%, Cauchy $P = 2.47 \times 10^{-7}$) (Supplementary Table 14). This association did not replicate in the deCODE or Genes & Health (G&H) datasets ($P > 0.05$; Supplementary Tables 14, 20 and 21).

**Novel genes implicated by gene-burden analyses.** Gene-based analyses identify 18 genes associated with testosterone-F and 20 genes with testosterone-M (Cauchy combination $P < 5 \times 10^{-6}$), of which ten have not previously been implicated through GWASs (Supplementary Note). Across 43 gene–trait pairs with Cauchy $P < 5 \times 10^{-6}$ in UKBB, 13 (30.2%) replicate at nominal significance ($P < 0.05$) and two (4.65%) at Bonferroni-adjusted significance ($P < 6.85 \times 10^{-4}$) in either the deCODE or G&H datasets with consistent directions of effect (Supplementary Tables 14, 20 and 21).

We replicated the testosterone-F lowering associations of rare damaging variation in the hydroxysteroid dehydrogenase enzymes *AKR1D1* (UKBB $P = 1.76 \times 10^{-8}$, deCODE $P = 1.08 \times 10^{-7}$, G&H $P = 0.862$) and *AKR1C3* (UKBB $P = 2.21 \times 10^{-9}$, deCODE $P = 1.12 \times 10^{-6}$, G&H $P = 8.75 \times 10^{-8}$) in external cohorts ($P < 6.85 \times 10^{-4}$, Bonferroni adjustment for 43 independent gene–trait pairs) (Supplementary Tables 14, 20 and 21). We report the first known association of *HSD11B1* with testosterone-F (burden test $P = 1.93 \times 10^{-6}$ when aggregated across missense variants with MAF <0.01%), with nominal replication in deCODE ($P = 0.028$); pathogenic variants in this gene are reported to cause hyperandrogenism due to cortisone reductase deficiency[49] (Supplementary Fig. 11). We also report the association of testosterone-M with *HSD17B2* (burden test $P = 1.33 \times 10^{-11}$ when aggregated across predicted loss-of-function (pLoF) variants with MAF <0.1%), which encodes the enzyme hydroxysteroid 17β-dehydrogenase 2 that regulates the biological potency of steroid hormones[50] (Supplementary Fig. 11 and Supplementary Table 14). The association of rare damaging variation in *HSD17B2* with lower testosterone nominally replicated in deCODE ($P = 2.22 \times 10^{-3}$) and G&H ($P = 0.0481$).

**Infertility risk linked to rare hormone-associated variants.** Two genes associated with testosterone in female UKBB participants were also associated with infertility risk ($P < 1.00 \times 10^{-3}$, Bonferroni adjustment for 50 unique genes): *TRIM4* (F-ALL, burden test OR 1.03 (1.01–1.05), $P = 4.05 \times 10^{-4}$ across all variants with MAF <0.1%) and *CYP3A43* (F-EXCL, burden test OR 1.02 (1.01–1.03), $P = 4.84 \times 10^{-4}$ across all variants with MAF <1%). Neither gene has previously been implicated in infertility.

Finally, we identified 29 unique genes carrying rare variants (MAF <1%) associated with testosterone in male or female participants in the UKBB, a majority of which had the same direction of effect in our EUR-ancestry GWAS meta-analyses excluding UKBB participants (Supplementary Table 15 and Supplementary Note). Nineteen of the 29 genes also contained nearby (±500 kb) common testosterone-associated variants from GWASs (MAF >1%), but at the majority (74%) of these loci, the effect of the rare variant was larger and remained upon conditioning on common variants ($P < 1 \times 10^{-7}$ after conditioning; Fig. 6a, Supplementary Table 15 and Supplementary Note).

The 11 novel testosterone associations included a female testosterone-lowering missense variant in *STAG3* (chr7:100204708:C:T, $\beta = -0.284$, $P = 2.31 \times 10^{-8}$). *STAG3* is also associated with primary ovarian insufficiency in women[51], and lack of *Stag3* results in female infertility through the absence of oocytes in knockout mouse models[23]. We did not find a significant association between the *STAG3* variant and female infertility in UKBB ($P > 0.05$). However, we observed increased risk of idiopathic infertility in women carrying a novel testosterone-lowering variant in *GPC2* (chr7:100171569:G:A, F-EXCL OR 2.63 (1.40–4.92), $P = 1.25 \times 10^{-3}$) (Fig. 6b). *GPC2* is highly expressed in the testis, and *Gpc2*-knockout mouse models display reduced adrenal gland size[23]. The gene has not previously been reported to be associated with testosterone or infertility. Taken together, our results indicate a potential role for infertility driven by rare hormone-disrupting variants.

## Discussion

Our large-scale genetic investigation of infertility and related reproductive phenotypes in over 1.5 million individuals identified 22 genetic loci associated with female infertility, three with male infertility and novel variants associated with levels of the reproductive hormones FSH, LH, estradiol and total testosterone in men and women. Through rare-variant and gene-based analyses in the UKBB, we additionally identified 50 genes associated with testosterone levels, including the first reported hormone-associated variants in some members of the hydroxysteroid dehydrogenase enzyme family. Although there was evidence for distinct genetic architectures of infertility and

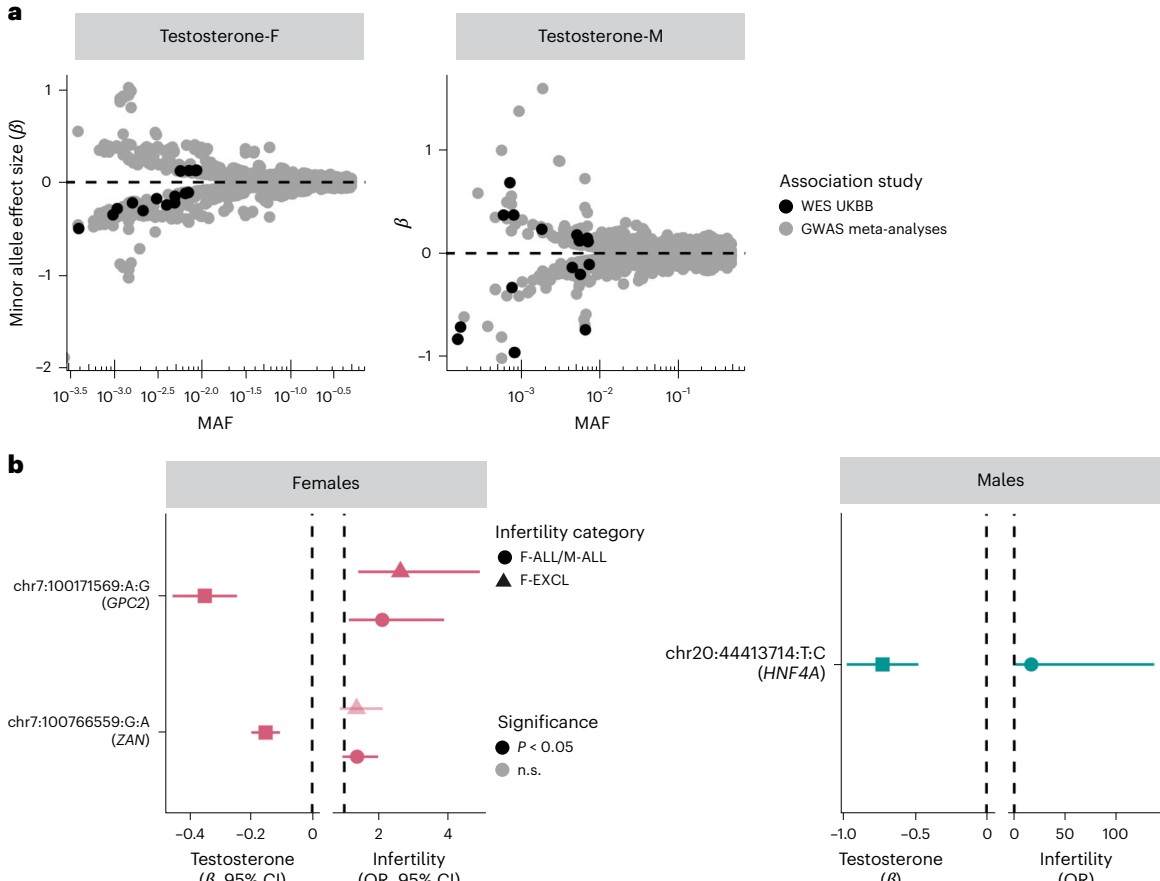

**Fig. 6 | Rare variants associated with testosterone and infertility in UKBB WES analyses. a**, The mean effect size versus allele frequency of genetic variants associated with total testosterone estimated using regression analyses. Variants discovered at genome-wide significance ($P < 5 \times 10^{-8}$) in GWAS meta-analyses ($n_{female} = 235,579$, $n_{male} = 235,096$) and exome-wide significance ($P < 1 \times 10^{-7}$) in the UKBB WES analyses ($n_{female} = 197,038$, $n_{male} = 197,340$) are plotted. The effect sizes are aligned to the minor allele, plotted against MAF on the log $x$ axis. **b**, The effects of testosterone-associated rare variants (chr:pos:minor allele:major allele) on infertility in females (left: $n$ cases/controls for F-ALL = 3,746/260,413;

$n$ cases/controls for F-EXCL = 3,012/261,147) and males (right: $n$ cases/controls for M-ALL = 650/222,393) estimated using regression analyses. The effect sizes are aligned to the minor allele. Per gene, the variant with the lowest $P$ value of all variants that reach exome-wide significance in UKBB WES analyses for testosterone is displayed, for all variants with nominally significant effects on infertility. Effect sizes ($\beta$ and 95% CIs) for the variant effect on testosterone are to the left of each plot and effect sizes (ORs and 95% CIs) for the variant effect on infertility are to the right of each plot.

reproductive hormones, we showed that individual genes containing rare protein-coding variants associated with testosterone (*GPC2*, *CYP3A43* and *TRIM4*) were also associated with higher risk of infertility in the UKBB.

Previous efforts to catalog the genome-wide architecture of infertility have relied on proxy measures such as childlessness and number of children ever born[10,11], which may be confounded by behavioral, socioeconomic and lifestyle factors. While we found modest genetic correlation between female infertility and age at first sexual intercourse (−18.8%), indicating that the latter captures some shared biology with fertility, our meta-analyses prioritize novel genes with putative roles in male and female gonads, such as *TRHR* for ovarian insufficiency and *ENO4* for sperm motility, those responsible for development of the endocrine system, such as *UMODL1*, and pharmacogenetic interactions, such as the association of *SLC47A2* with female infertility, potentially mediated by response to metformin.

The strong genetic correlation of 71% between idiopathic infertility and endometriosis may indicate that some proportion of idiopathic cases are due to underdiagnosis of endometriosis, whose early treatment may prevent future infertility[52]. Our subtype-specific analyses highlight the value in dissecting heterogeneous causes of infertility. For example, PCOS is a heritable cause of up to 80% of anovulatory

infertility cases that may be treated through induced ovulation[53]. However, only three of eight loci for anovulatory infertility colocalize with known PCOS signals, the genetic correlation between these traits is only 40% and the majority (88%) of causal variants for anovulatory infertility are not shared with PCOS. These results suggest that other hypothalamic–pituitary–ovarian disorders, endocrinopathies (hypothyroidism, hyperprolactinemia and so on) and ovarian insufficiency may also contribute significantly to the genetic etiology of anovulatory infertility, and require treatments different from those for PCOS-associated infertility[54]. Weight loss for overweight patients is often recommended as beneficial for fertility[55,56], but we did not find substantial genetic correlation between obesity and infertility. Our findings add genetic support to evidence from randomized controlled trials demonstrating limited fertility benefits from short-term weight loss in overweight and obese women[57]. Instead, we observed bidirectional causal relationships between abdominal obesity and anovulatory infertility, suggesting physiological feedback mechanisms whose complex interplay requires deeper study.

Our results indicate that balancing selection and recent positive selection at pleiotropic loci may explain the persistence of genetic factors for infertility. For example, the *EBAG9* locus associated with female infertility is under directional selection, perhaps because *EBAG9*

plays a role in the adaptive immune memory response to infection[58]. A complementary role for *EBAG9* may be in the placenta during early pregnancy, where reduction of *EBAG9* levels is associated with inappropriate activation of the maternal immune system and results in fetal rejection[15].

We employed a broad search strategy to maximize sample sizes for cases of infertility and reproductive hormone levels in our meta-analyses, which has its limitations. Diagnostic criteria for infertility vary by country and have changed over time[1], which may explain the wide spread in the prevalence of infertility across cohorts. Reproductive hormone values in this study were assayed using different methodologies and at different ages and stages of the menstrual cycle in women. A majority of samples in our study were derived from the UKBB and measured during and postmenopause (ages 40–69 years), whereas infertility occurs premenopause, so we urge caution in interpreting the lack of correlation between these traits. Although we were able to adjust for covariates such as age, which can account for some of the effect of menopause on hormone levels, we did not have the data granularity to account for hormonal fluctuations during the menstrual cycle and pregnancy.

In this comprehensive large-scale investigation of the genetic determinants of infertility and reproductive hormones across men and women, we identified several genes associated with infertility and analyzed their effects on reproductive disease and selection pressures. We did not find evidence that reproductive hormone dysregulation and obesity are strongly correlated with infertility at the population level, but instead nominate individual hormone-associated genes with effects on fertility. Other genetic and non-genetic avenues must be explored to treat complex and heterogeneous fertility disorders that impact the physical, emotional and financial well-being of millions of individuals across the globe.

## Online content

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

**Samvida S. Venkatesh** [1,2] ✉, **Laura B. L. Wittemans**[3,4], **Duncan S. Palmer** [1,5], **Nikolas A. Baya** [1,2], **Teresa Ferreira**[1], **Barney Hill**[1,5], **Frederik Heymann Lassen**[1,2], **Melody J. Parker** [1,6], **Saskia Reibe**[1,5], **Ahmed Elhakeem**[7,8], **Karina Banasik**[9,10], **Mie T. Bruun** [11], **Christian Erikstrup** [12,13], **Bitten Aagard Jensen**[14], **Anders Juul** [15,16], **Christina Mikkelsen** [17,18], **Henriette S. Nielsen** [15,19], **Sisse R. Ostrowski** [15,17], **Ole B. Pedersen** [15,20], **Palle Duun Rohde** [21], **Erik Sørensen** [17], **Henrik Ullum**[22], **David Westergaard**[9,10], **Asgeir Haraldsson** [23,24], **Hilma Holm** [25], **Ingileif Jonsdottir** [23,25], **Isleifur Olafsson**[26], **Thora Steingrimsdottir**[23,27], **Valgerdur Steinthorsdottir** [25], **Gudmar Thorleifsson** [25], **Jessica Figueredo**[28], **Minna K. Karjalainen** [29,30,31], **Anu Pasanen**[32], **Benjamin M. Jacobs** [33], **Georgios Kalantzis** [34], **Nikki Hubers** [35,36,37], **Genes & Health Research Team***, **Estonian Biobank Research Team***, **Estonian Health Informatics Research Team***, **DBDS Genomic Consortium***, **FinnGen***, **Margaret Lippincott** [38,39], **Abigail Fraser**[7,8],

Article

Deborah A. Lawlor ⓘ [7,8], Nicholas J. Timpson ⓘ [7,8], Mette Nyegaard ⓘ [21,22], Kari Stefansson ⓘ [23,25], Reedik Magi ⓘ [28], Hannele Laivuori ⓘ [29,40,41,42], David A. van Heel ⓘ [43], Dorret I. Boomsma ⓘ [36,37], Ravikumar Balasubramanian ⓘ [38,39], Stephanie B. Seminara[38,39], Yee-Ming Chan ⓘ [39,44], Triin Laisk ⓘ [28] & Cecilia M. Lindgren ⓘ [1,2,4,45] ✉

[1]Big Data Institute, Li Ka Shing Centre for Health Information and Discovery, University of Oxford, Oxford, UK. [2]Wellcome Centre for Human Genetics, Nuffield Department of Medicine, University of Oxford, Oxford, UK. [3]Novo Nordisk Research Centre Oxford, Oxford, UK. [4]Nuffield Department of Women's and Reproductive Health, Medical Sciences Division, University of Oxford, Oxford, UK. [5]Nuffield Department of Population Health, Medical Sciences Division, University of Oxford, Oxford, UK. [6]Nuffield Department of Clinical Medicine, University of Oxford, John Radcliffe Hospital, Oxford, UK. [7]MRC Integrative Epidemiology Unit at the University of Bristol, Bristol, UK. [8]Population Health Science, Bristol Medical School, University of Bristol, Bristol, UK. [9]Novo Nordisk Foundation Center for Protein Research, University of Copenhagen, Copenhagen, Denmark. [10]Department of Obstetrics and Gynecology, Copenhagen University Hospital, Hvidovre, Copenhagen, Denmark. [11]Department of Clinical Immunology, Odense University Hospital, Odense, Denmark. [12]Department of Clinical Immunology, Aarhus University Hospital, Aarhus, Denmark. [13]Department of Clinical Medicine, Health, Aarhus University, Aarhus, Denmark. [14]Department of Clinical Immunology, Aalborg University Hospital, Aalborg, Denmark. [15]Department of Clinical Medicine, Faculty of Health and Medical Sciences, University of Copenhagen, Copenhagen, Denmark. [16]Department of Growth and Reproduction, Copenhagen University Hospital–Rigshospitalet, Copenhagen, Denmark. [17]Department of Clinical Immunology, Copenhagen University Hospital–Rigshospitalet, Copenhagen, Denmark. [18]Novo Nordisk Foundation Center for Basic Metabolic Research, Faculty of Health and Medical Science, Copenhagen University, Copenhagen, Denmark. [19]Department of Obstetrics and Gynecology, The Fertility Clinic, Hvidovre University Hospital, Copenhagen, Denmark. [20]Department of Clinical Immunology, Zealand University Hospital–Køge, Køge, Denmark. [21]Genomic Medicine, Department of Health Science and Technology, Aalborg University, Aalborg, Denmark. [22]Statens Serum Institut, Copenhagen, Denmark. [23]Faculty of Medicine, University of Iceland, Reykjavik, Iceland. [24]Children's Hospital Iceland, Landspitali University Hospital, Reykjavik, Iceland. [25]deCODE genetics/Amgen Inc., Reykjavik, Iceland. [26]Department of Clinical Biochemistry, Landspitali University Hospital, Reykjavik, Iceland. [27]Department of Obstetrics and Gynecology, Landspitali University Hospital, Reykjavik, Iceland. [28]Estonian Genome Centre, Institute of Genomics, University of Tartu, Tartu, Estonia. [29]Institute for Molecular Medicine Finland, Helsinki Institute of Life Science, University of Helsinki, Helsinki, Finland. [30]Research Unit of Population Health, Faculty of Medicine, University of Oulu, Oulu, Finland. [31]Northern Finland Birth Cohorts, Arctic Biobank, Infrastructure for Population Studies, Faculty of Medicine, University of Oulu, Oulu, Finland. [32]Research Unit of Clinical Medicine, Medical Research Center Oulu, University of Oulu, and Department of Children and Adolescents, Oulu University Hospital, Oulu, Finland. [33]Centre for Preventive Neurology, Wolfson Institute of Population Health, Queen Mary University London, London, UK. [34]Wellcome Sanger Institute, Wellcome Genome Campus, Hinxton, UK. [35]Department of Biological Psychology, Netherlands Twin Register, Vrije Universiteit, Amsterdam, The Netherlands. [36]Amsterdam Reproduction and Development Institute, Amsterdam, The Netherlands. [37]Department of Complex Trait Genetics, Center for Neurogenomics and Cognitive Research, Vrije Universiteit, Amsterdam, The Netherlands. [38]Reproductive Endocrine Unit, Massachusetts General Hospital, Boston, MA, USA. [39]Harvard Medical School, Boston, MA, USA. [40]Medical and Clinical Genetics, University of Helsinki and Helsinki University Hospital, Helsinki, Finland. [41]Department of Obstetrics and Gynecology, Tampere University Hospital, The Wellbeing Services County of Pirkanmaa, Tampere, Finland. [42]Center for Child, Adolescent, and Maternal Health Research, Faculty of Medicine and Health Technology, Tampere University, Tampere, Finland. [43]Blizard Institute, Queen Mary University London, London, UK. [44]Division of Endocrinology, Department of Pediatrics, Boston Children's Hospital, Boston, MA, USA. [45]Broad Institute of Harvard and MIT, Cambridge, MA, USA. *Lists of authors and their affiliations appear at the end of the paper. ✉e-mail: samvida.venkatesh@gmail.com; cecilia.lindgren@bdi.ox.ac.uk

### Genes & Health Research Team

David A. van Heel[43]

### Estonian Biobank Research Team

Triin Laisk[28], Reedik Mägi[28], Andres Metspalu[28], Lili Milani[28,46], Tõnu Esko[28], Mari Nelis[28] & Georgi Hudjashov[28]

[46]Estonian Biobank, Institute of Genomics, University of Tartu, Tartu, Estonia.

### Estonian Health Informatics Research Team

Raivo Kolde[47], Sven Laur[47,48], Sulev Reisberg[47,48] & Jaak Vilo[47,48]

[47]Institute of Computer Science, University of Tartu, Tartu, Estonia. [48]STACC, Tartu, Estonia.

### DBDS Genomic Consortium

Karina Banasik[9,10], Mie T. Bruun[11], Christian Erikstrup[12,13], Bitten Aagard Jensen[14], Christina Mikkelsen[17,18], Henriette S. Nielsen[15,19], Sisse R. Ostrowski[15,17], Ole B. Pedersen[15,20], Palle Duun Rohde[21], Erik Sørensen[17], Henrik Ullum[22], David Westergaard[9,10], Mette Nyegaard[21,22] & Kari Stefansson[23,25]

### FinnGen

Minna K. Karjalainen[29,31] & Hannele Laivuori[29,40,41,42]

## Methods

This research complies with all relevant ethical regulations. Each contributing study was approved by its respective board/committee as detailed in the Supplementary Note.

### Study populations and phenotype identification

**Binary traits (infertility).** Cases were identified in UKBB, Copenhagen Hospital Biobank and Danish Blood Donor Study, deCODE, EstBB, FinnGen and G&H (Supplementary Methods). We defined five categories of female infertility: F-ALL, F-ANOV, F-ANAT (including tubal, uterine and cervical origins), F-EXCL and F-INCL, and male infertility of all causes (M-ALL). Cases were identified through self-report (F-ALL, F-EXCL and M-ALL) and through primary- and secondary-care codes (Supplementary Table 1). Within each subtype, sex-matched controls were defined as individuals not identified as cases for that subtype. We additionally included publicly available multi-ancestry summary statistics from the Million Veteran Program (MVP) in meta-analyses of F-ALL (PheCode 626.8) and M-ALL (PheCode 609), downloaded from dbGaP[59].

**Quantitative traits (reproductive hormones).** Hormones were included from UKBB, Avon Longitudinal Study of Parents and Children (ALSPAC), deCODE, EstBB and G&H (Supplementary Note). We extracted measurements of FSH, LH, estradiol, progesterone and testosterone from biobank assessment centers or primary- and secondary-care records (Supplementary Table 16). If repeated measurements were available for an individual, we retained the recorded hormone value closest to the individual's median hormone value over time. Each hormone was regressed on age, age$^2$ and cohort-specific covariates specified below; the residuals from this regression were rank-based inverse normally transformed before GWAS.

### Meta-analysis of GWAS summary statistics

**Genome-wide association testing.** Association analyses were performed separately within each ancestry and sex stratum for all strata with at least 100 cases (binary traits) or 1,000 individuals (quantitative traits). For binary traits, each variant passing quality control (QC) was tested for association under an additive model using REGENIE[60] or Scalable and Accurate Implementation of GEneralized mixed model (SAIGE)[61], with adjustments for age, age$^2$ and cohort-specific covariates, with the Firth correction applied to control for inflation at rare variants and traits with low case–control ratios[60,61]. For quantitative traits, the rank-based inverse normally transformed hormone value was tested for association under an additive model using REGENIE[60] or SAIGE[61], with adjustments for cohort-specific genetic covariates. Any deviations from this GWAS protocol are noted in the Supplementary Note.

**Meta-analysis.** Before meta-analysis, summary statistics from all studies underwent thorough QC to retain variants that met the following criteria: (1) on the autosomes or X chromosome; (2) with imputation information score >0.8 (where available); (3) bi-allelic variants with A, C, G, T alleles; (4) with s.e.m. <10 and $P$ values in (0,1); and (5) without duplicate entries. Fixed-effects inverse-variance-weighted meta-analysis was performed using METAL[62]. We report results from EUR-ancestry and all-ancestry meta-analyses for each trait. Genome-wide significance was established at $P < 5 \times 10^{-8}$.

**Identification of lead variants.** Distance-based pruning was used to identify lead variants as the SNP with the lowest $P$ value within each 1 Mb window at all loci with at least one genome-wide significant variant with $P < 5 \times 10^{-8}$. Lead hormone-associated variants were classified as novel or previously reported (Supplementary Note).

### SNP-based heritability

The following analyses, which rely on population-specific LD patterns, were restricted to EUR-ancestry summary statistics with precomputed LD scores based on EUR-ancestry individuals in the 1000 Genomes dataset[63], restricted to HapMap3 SNPs[64]. We estimated the SNP-based heritability ($h_G^2$) of a trait from GWAS summary statistics using LD-score regression as implemented in the LDSC software[65]. For infertility traits, the observed-scale heritability ($h_{obs}^2$) was converted to liability-scale heritability ($h_{liab}^2$), which accounts for the disease prevalence in the sample ($k$) and population ($K$), under the assumption that sample prevalence equals the population prevalence[66].

### Genetic correlations

**Genome-wide genetic correlations.** LDSC was used to estimate genetic correlations between infertility traits, hormone levels and a collection of other phenotypes in the UKBB in EUR-ancestry individuals. To simplify computation of $r_g$ across a large number of traits, we used an extension of the LDSC software that allows for simultaneous estimation of multiple genetic correlations[67] (Supplementary Note).

**Local genetic correlations.** We applied LAVA v1.8.0 to assess bivariate correlations and perform multivariate regressions at local genomic regions[68], using 1000 Genomes reference genotype data. Genomic loci that were created by partitioning the genome into 2,495 blocks of approximately 1 Mb each, while minimizing LD between blocks, were downloaded from ref. [69]. As we quantified sample overlap between traits using cross-trait LDSC, the phenotypes assessed were restricted to those with significant genome-wide heritability ($Z > 4$): female infertility (F-ALL, F-ANOV and F-INCL), five reproductive conditions (endometriosis, PCOS, heavy menstrual bleeding, uterine fibroids and dizygotic twinning), four hormones (FSH-F, testosterone-F, TSH and AMH) and five obesity-related traits (BMI, body fat percentage, waist circumference, hip circumference and comparative body size at age 10 years). The target phenotype (female infertility: F-ALL, F-ANOV or F-INCL) was tested against all other traits.

At each genomic block, we filtered to phenotypes with sufficient local heritability ($P < 2.00 \times 10^{-5}$, family-wise error rate (FWER) controlled at 5% across 2,495 regions using the Bonferroni method, as recommended by the developers of LAVA[68]), resulting in 2,618 pairwise bivariate $r_g$ tests. At any regions where female infertility was correlated with multiple traits, we performed multiple regression to assess the independent predictive power of each trait. If traits were collinear in the region (local $r_g > 0.9$), the trait with the greatest local $r_g$ with infertility was retained.

**Polygenic overlap.** We estimated polygenic overlap, irrespective of genetic correlation, using the causal mixture model MiXeR v1.3 (ref. [32]) applied to GWAS summary statistics for female infertility (F-ALL, F-ANOV and F-INCL), five reproductive conditions (endometriosis, PCOS, heavy menstrual bleeding, uterine fibroids and dizygotic twinning), four reproductive hormones (FSH-F, testosterone-F, TSH and AMH) and five obesity-related traits (BMI, body fat percentage, waist circumference, hip circumference and comparative body size at age 10 years). For bivariate tests, the target phenotype (female infertility: F-ALL, F-ANOV or F-INCL) was tested against all other traits. GWAS summary statistics were formatted as for LDSC and LD structure was estimated using the 1000 Genomes project, as described previously[32]. Both univariate and bivariate models were fitted in two steps to ensure robust convergence, where the first 'fast model' estimates a set of parameters that are then constrained in the second 'full model'. We only report estimates and s.e.m. for parameters from the 'full model', with the exception of Akaike information criterion (AIC) and Bayesian information criterion (BIC) from the first model, which are used to compare the univariate MiXeR model with LDSC regression, as recommended by the developers of MiXeR[32]. The number of causal variants is estimated from the fraction of total causal variants that cumulatively account for 90% of heritability in each component. Genome-wide genetic correlation estimates from MiXeR and LDSC were comparable as we used

the same set of SNPs for both analyses; we calculated the P value for heterogeneity between these estimates using the z score for difference.

## MR

Analyses were all performed with summary statistics from EUR-ancestry GWASs, using the TwoSampleMR v0.5.7 package[70]. Details of instrument construction, sensitivity analyses and MR methods are in the Supplementary Note.

## Colocalization

The following analyses were all performed with summary statistics from EUR-ancestry GWASs, using the Bayesian framework implemented in the coloc v5.1.0 package[71] under a single causal variant assumption[72]. Parameters for colocalization are provided in the Supplementary Note. We tested for colocalization between each female infertility category and each female-specific hormone trait (FSH, LH, estradiol and testosterone) at all genetic loci associated with at least one of the pair of traits tested. The single male infertility locus with common variants (MAF >1%) in the EUR-ancestry analysis did not contain enough significant associations (only 12 common variants with $P < 1 \times 10^{-6}$) for colocalization analyses. As we noticed that some lead variants for female infertility had previously been reported as associated with endometriosis and PCOS, we estimated the PP of colocalization of genetic signals between each category of female infertility and each of these two reproductive disorders. EUR-ancestry summary statistics for endometriosis and PCOS were obtained as described in 'Genetic correlations' section above. We assessed colocalization of genetic signals for female infertility with eQTLs for all proximal genes with transcription start sites (TSSs) within 1 Mb of an infertility lead variant. Publicly available eQTL data was downloaded from the GTEx project[73].

## Tissue and cell-type prioritization

We estimated the polygenic contributions of genes with tissue-specific expression profiles to the heritability of infertility and hormones using stratified LD-score regression (partitioned heritability analyses)[65]. We restricted these analyses to traits with highly significant heritability in EUR-ancestry analyses (Z > 7) (F-ALL, testosterone-F and testosterone-M), as recommended by the developers[74].

Gene sets and LD scores for 205 tissues and cell types from the GTEx project database[73] and the Franke laboratory single-cell database[75] were downloaded from ref. [76]. We established tissue-wide significance at $-\log_{10}(P) > 2.75$, which corresponds to a false discovery rate (FDR) <5%.

As the ovary, a reproductive tissue of interest, is not well characterized in the GTEx project, we constructed annotation-specific LD scores for ovarian cell types from two publicly available single-cell RNA sequencing datasets (Supplementary Note).

## Overlaps with genetic regions under selection

To avoid confounding by population stratification, selection look-ups were restricted to GWAS summary statistics from EUR-ancestry individuals. We assessed selection in three forms: (1) loci under directional selection, following guidelines described by Mathieson et al.[11], (2) recent trait-wide directional selection using SDSs, following the protocol outlined by Field et al.[39] and (3) balancing selection using standardized BetaScan2 scores[40] (Supplementary Note).

## WES analyses

**Exome sequencing QC.** *QC outline*. We first considered an initial set of 'high-quality' variants to evaluate the mean call rate and depth of coverage for each sample. We then ran a sample and variant level prefiltering step and calculated sample-level QC metrics. Using these metrics, we removed sample outliers based on median absolute deviation thresholds and excluded sites that did not pass variant QC according to Karczewski et al.[77]. We then applied a genotype-level filter using

genotype quality, depth and heterozygote allele balance. The resultant high-quality EUR call set consisted of 402,375 samples and 25,229,669 variants. For details, see Supplementary Note.

**Variant annotation.** We annotated variants using Variant Effect Predictor (VEP) v105 (corresponding to gencode v39)[78] with the LOFTEE v1.04_GRCh38 (ref. [79]) and dbNSFP[80] plugins, annotating variants with CADD v1.6 (ref. [81]) and REVEL using dbNSFP4.3 (ref. [82]) and loss of function confidence using LOFTEE. Complete instructions and code for this step are provided in our VEP_105_LOFTEE repository[83], which contains a docker/singularity container to ensure reproducibility of annotations. We then ran SpliceAI v1.3 (ref. [84]) using the gencode v39 gene annotation file to ensure alignment between VEP and SpliceAI transcript annotations. We defined 'canonical' transcripts to be used for variant-specific annotations as follows: set MANE Select[85] as the canonical, where available, and if a MANE Select transcript is not present, set canonical and restrict to protein-coding genes. For VEP 105, this is equivalent to selecting the 'canonical' transcript in protein-coding genes. Then, using the collection of missense, pLoF, splice metrics and annotations of variant consequence on the 'canonical' transcript, we determined a set of variant categories for gene-based testing (Supplementary Note).

Variant counts and average allele counts for each annotation, split by population label and binned by MAF are displayed in Supplementary Figs. 13 and 14, respectively.

**Genetic association testing.** We carried out rare-variant genetic association testing in the EUR-ancestry subset of the UKBB using SAIGE[61], a mixed model framework that accounts for sample relatedness and case–control imbalance through a saddle-point approximation in binary traits. All rare-variant analysis was carried out on the UKBB Research Analysis Platform using SAIGE version wzhou88/saige:1.1.9 (ref. [61]). In the sex-combined analyses, we account for age, sex, age², age × sex, age² × sex and the first 10 genetic principal components as fixed effects; and age, age² and the first 10 principal components in sex-specific analyses. All continuous traits were inverse rank normalized before association testing.

For SAIGE step 0, we constructed a genetic relatedness matrix (GRM) using the UKBB genotyping array data. We LD pruned the genotyped data using PLINK (–indep-pairwise 50 5 0.05)[86] and created a sparse GRM using 5,000 randomly selected markers, with relatedness cutoff of 0.05, using the createSparseGRM.R function within SAIGE. To generate a variance ratio file for subsequent steps in SAIGE, we extracted 1,000 variants each with minor allele count (MAC) <20 and MAC >20, and combined these markers to define a PLINK file for the variance ratio determination.

In SAIGE step 1 for each trait, the curated phenotype data and sparse GRM were used to fit a null model with no genetic contribution. All parameters were set at the defaults in SAIGE, except --relatednessCutoff 0.05, --useSparseGRMtoFitNULL TRUE and --isCateVarianceRatio TRUE. Tolerance for fitting the null generalized linear mixed model was set to 0.00001.

*Rare-variant and gene-based testing*. Following null model fitting, we carried out variant and gene-based testing in SAIGE step 2 using the variant categories described above, with the --is_single_in_group Test TRUE flag. All other parameters were set to default, except --maxMAF_in_groupTest=0.0001,0.001,0.01, --is_Firth_beta TRUE, --pCutofforFirth=0.1 and --is_fastTest TRUE. We included the following collection of group tests, using the annotations defined above (see 'Variant annotation' section).

- High confidence pLoF
- Damaging missense/protein altering
- Other missense/protein altering

- Synonymous
- High confidence pLoF or damaging missense/protein altering
- High confidence pLoF or damaging missense/protein altering or other missense/protein altering or synonymous

We then carried out Cauchy combination tests[87] across these annotations for each gene.

**Conditioning on nearby common variants.** We conditioned rare variants that reached significance in exome-wide analyses ($P < 1 \times 10^{-7}$, Bonferroni adjustment for ~500,000 independent variants) on nearby common lead variants (±500 kb with MAF >1%), identified from the GWAS meta-analyses reported in this manuscript. The identified lead SNPs were included as covariates in SAIGE variant-based tests step 2.

**Replication of exome-sequencing analyses.** QC, variant annotation and genetic association testing at the gene level in the G&H whole-exome sequencing (WES) data were performed identically to UKBB, with any deviations noted below. The total sample size available was 44,028 (24,444 females and 19,584 males) with phenotype-specific sample sizes provided in Supplementary Tables 2 and 9. Sample sizes for WES analyses in G&H differ from sample sizes for GWASs because (1) further samples were added during the period between analyses and (2) not all G&H samples with genotyping have been exome sequenced as yet. A total of 4,723,926 variants passed QC. Genetic association analysis covariates were age, sex, age$^2$ and the first 20 genetic principal components as fixed effects; and age, age$^2$ and the first 20 principal components in sex-specific analyses.

Variant annotation, gene-burden models and association analyses performed in deCODE are described in the Supplementary Note.

**Reporting summary**

Further information on research design is available in the Nature Portfolio Reporting Summary linked to this article.

## Data availability

Cohorts may be contacted individually for access to raw data. Meta-analysis summary statistics for all phenotypes are available through the GWAS Catalog study accession nos. GCST90483463 to GCST90483502.

## Code availability

All code used in this study is available via GitHub at https://github.com/lindgrengroup/infertility_hormones and via Zenodo at https://doi.org/10.5281/zenodo.14528258 (ref. 88).

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

## Acknowledgements

We thank P. Palamara and Z. Kutalik for helpful discussions. We are grateful to the participants of all cohorts. This research has partly

been conducted using the UKBB Resource under application no. 11867. G&H is/has recently been core-funded by Wellcome (WT102627, WT210561), the Medical Research Council (UK) (M009017, MR/X009777/1 and MR/X009920/1), Higher Education Funding Council for England Catalyst, Barts Charity (845/1796), Health Data Research UK (for London substantive site) and research delivery support from the NHS National Institute for Health Research Clinical Research Network (North Thames). G&H is/has recently been funded by Alnylam Pharmaceuticals, Genomics PLC; and a Life Sciences Industry Consortium of Astra Zeneca PLC, Bristol-Myers Squibb Company, GlaxoSmithKline Research and Development Limited, Maze Therapeutics Inc, Merck Sharp & Dohme LLC, Novo Nordisk A/S, Pfizer Inc, Takeda Development Centre Americas Inc. We thank Social Action for Health, Centre of The Cell, members of our Community Advisory Group and staff who have recruited and collected data from volunteers. We thank the NIHR National Biosample Centre (UK Biocentre), the Social Genetic and Developmental Psychiatry Centre (King's College London), Wellcome Sanger Institute and Broad Institute for sample processing, genotyping, sequencing and variant annotation. We thank Barts Health NHS Trust, NHS Clinical Commissioning Groups (City and Hackney, Waltham Forest, Tower Hamlets, Newham, Redbridge, Havering, Barking and Dagenham), East London NHS Foundation Trust, Bradford Teaching Hospitals NHS Foundation Trust, Public Health England (especially D. Wyllie), Discovery Data Service/Endeavour Health Charitable Trust (especially D. Stables), Voror Health Technologies Ltd (especially S. Don) and NHS England (for what was NHS Digital) for GDPR-compliant data sharing backed by individual written informed consent. Most of all, we thank all of the volunteers participating in G&H. This study was funded by the European Union through the European Regional Development Fund project no. 2014-2020.4.01.15-0012 GENTRANSMED. Data analysis was carried out in part in the High-Performance Computing Center of University of Tartu. This research has partly been conducted using the ALSPAC resource under project no. B4568. The activities of the EstBB are regulated by the Human Genes Research Act, which was adopted in 2000 specifically for the operations of the EstBB. Individual-level data analysis in the EstBB was carried out under ethical approval 1.1-12/624 from the Estonian Committee on Bioethics and Human Research (Estonian Ministry of Social Affairs), using data according to release application 3-10/GI/10790 from the EstBB. This study was funded by the Estonian Research Council grants PRG1911 and by the Ministry of Education and Research Centres of Excellence grant TK214 Centre of Excellence for Personalised Medicine. We are extremely grateful to all the families who took part in this study, the midwives for their help in recruiting them and the whole ALSPAC team, which includes interviewers, computer and laboratory technicians, clerical workers, research scientists, volunteers, managers, receptionists and nurses. The UK Medical Research Council and Wellcome (grant ref. 217065/Z/19/Z) and the University of Bristol provide core support for ALSPAC. A comprehensive list of grants funding is available on the ALSPAC website (http://www.bristol.ac.uk/alspac/external/documents/grant-acknowledgements.pdf). Genome-wide genotyping data was generated by Sample Logistics and Genotyping Facilities at Wellcome Sanger Institute and LabCorp (Laboratory Corporation of America) using support from 23andMe. We acknowledge the contribution of the participants in the deCODE study. Patients and control subjects in FinnGen provided informed consent for biobank research, based on the Finnish Biobank Act. Alternatively, separate research cohorts, collected prior to when the Finnish Biobank Act came into effect (in September 2013) and start of FinnGen (August 2017), were collected based on study-specific consents and later transferred to the Finnish biobanks after approval by Fimea (Finnish Medicines Agency), the National Supervisory Authority for Welfare and Health. Recruitment protocols followed the biobank protocols approved by Fimea. The Coordinating Ethics Committee of the Hospital District of Helsinki and Uusimaa (HUS) statement number for the FinnGen study is Nr HUS/990/2017. The FinnGen study is approved by Finnish Institute for Health and Welfare (permit nos. THL/2031/6.02.00/2017, THL/1101/5.05.00/2017, THL/341/6.02.00/2018, THL/2222/6.02.00/2018, THL/283/6.02.00/2019, THL/1721/5.05.00/2019 and THL/1524/5.05.00/2020), Digital and population data service agency (permit nos. VRK43431/2017-3, VRK/6909/2018-3 and VRK/4415/2019-3), the Social Insurance Institution (permit nos. KELA 58/522/2017, KELA 131/522/2018, KELA 70/522/2019, KELA 98/522/2019, KELA 134/522/2019, KELA 138/522/2019, KELA 2/522/2020 and KELA 16/522/2020), Findata (permit numbers THL/2364/14.02/2020, THL/4055/14.06.00/2020, THL/3433/14.06.00/2020, THL/4432/14.06/2020, THL/5189/14.06/2020, THL/5894/14.06.00/2020, THL/6619/14.06.00/2020, THL/209/14.06.00/2021, THL/688/14.06.00/2021, THL/1284/14.06.00/2021, THL/1965/14.06.00/2021, THL/5546/14.02.00/2020, THL/2658/14.06.00/2021 and THL/4235/14.06.00/2021), Statistics Finland (permit nos. TK-53-1041-17 and TK/143/07.03.00/2020 (earlier TK-53-90-20) TK/1735/07.03.00/2021, TK/3112/07.03.00/2021) and Finnish Registry for Kidney Diseases permission/extract from the meeting minutes on 4 July 2019. The Biobank Access Decisions for FinnGen samples and data utilized in FinnGen Data Freeze 10 include THL Biobank BB2017_55, BB2017_111, BB2018_19, BB_2018_34, BB_2018_67, BB2018_71, BB2019_7, BB2019_8, BB2019_26, BB2020_1, BB2021_65, Finnish Red Cross Blood Service Biobank 7.12.2017, Helsinki Biobank HUS/359/2017, HUS/248/2020, HUS/150/2022 § 12, §13, §14, §15, §16, §17, §18, and §23, Auria Biobank AB17-5154 and amendment no. 1 (17 August 2020) and amendments BB_2021-0140, BB_2021-0156 (26 August 2021, 2 February 2022), BB_2021-0169, BB_2021-0179, BB_2021-0161, AB20-5926 and amendment no. 1 (23 April 2020) and its modification (22 September 2021), Biobank Borealis of Northern Finland_2017_1013, 2021_5010, 2021_5018, 2021_5015, 2021_5023, 2021_5017, 2022_6001, Biobank of Eastern Finland 1186/2018 and amendment 22 § /2020, 53§/2021, 13§/2022, 14§/2022, 15§/2022, Finnish Clinical Biobank Tampere MH0004 and amendments (21.02.2020 and 06.10.2020), §8/2021, §9/2022, §10/2022, §12/2022, §20/2022, §21/2022, §22/2022, §23/2022, Central Finland Biobank 1-2017, and Terveystalo Biobank STB 2018001 and amendment 25 August 2020, Finnish Hematological Registry and Clinical Biobank decision 18 June 2021, Arctic biobank P0844: ARC_2021_1001. We acknowledge the participants and investigators of FinnGen study. The FinnGen project is funded by two grants from Business Finland (HUS 4685/31/2016 and UH 4386/31/2016) and the following industry partners: AbbVie Inc., Astra Zeneca UK Ltd, Biogen MA Inc., Bristol-Myers Squibb (and Celgene Corporation & Celgene International II Sàrl), Genentech Inc., Merck Sharp & Dohme LCC, Pfizer Inc., GlaxoSmithKline Intellectual Property Development Ltd., Sanofi US Services Inc., Maze Therapeutics Inc., Janssen Biotech Inc, Novartis Pharma AG and Boehringer Ingelheim International GmbH. We acknowledge the following biobanks for delivering biobank samples to FinnGen: Auria Biobank (www.auria.fi/biopankki), THL Biobank (www.thl.fi/biobank), Helsinki Biobank (www.helsinginbiopankki.fi), Biobank Borealis of Northern Finland (https://www.ppshp.fi/Tutkimus-ja-opetus/Biopankki/Pages/Biobank Borealis-briefly-in-English.aspx), Finnish Clinical Biobank Tampere (www.tays.fi/en-US/Research_and_development/Finnish_Clinical_Biobank_Tampere), Biobank of Eastern Finland (www.ita-suomenbiopankki.fi/en), Central Finland Biobank (www.ksshp.fi/fi-FI/Potilaalle/Biopankki), Finnish Red Cross Blood Service Biobank (www.veripalvelu.fi/verenluovutus/biopankkitoiminta), Terveystalo Biobank (www.terveystalo.com/fi/Yritystietoa/Terveystalo-Biopankki/Biopankki/) and Arctic Biobank (https://www.oulu.fi/en/university/faculties-and-units/faculty-medicine/northern-finland-birth-cohorts-and-arctic-biobank). All Finnish Biobanks are members of BBMRI.fi infrastructure (www.bbmri.fi). Finnish Biobank Cooperative–FINBB

(https://finbb.fi/) is the coordinator of BBMRI–ERIC operations in Finland. The Finnish biobank data can be accessed through the Fingenious services (https://site.fingenious.fi/en/) managed by FINBB. This study was based on the CHB reproduction protocol and DBDS (ethical approval NVK-1805807; NVK-1700407; SJ-740). S.S.V. was supported by the Rhodes Scholarships, Clarendon Fund, and the Medical Sciences Doctoral Training Centre at the University of Oxford. L.B.L.W. was supported by the Wellcome Trust. B.M.J. was funded by a Medical Research Council (MRC) Clinical Research Training Fellowship (CRTF) jointly supported by the UK MS Society (B.M.J.; grant reference: MR/V028766/1). A.P. is supported by Alma and K.A. Snellman Foundation. Contributions by A.E. and D.A.L. are supported by the UK Medical Research Council (MC_UU_00032/05) and the European Research Council under the European Union's Horizon 2020 research and innovation program (grant agreement no. 101021566). D.A.L. is also supported by the British Heart Foundation (CH/F/20/90003). G.K. is funded by Wellcome (grant no. 220540/Z/20/A, 'Wellcome Sanger Institute Quinquennial Review 2021–2026'). D.W., K.B. and H.S.N. acknowledge the Novo Nordisk Foundation (grant nos. NNF22OC0077221 and NNF23OC0087269, NNF17OC0027594 and NNF14CC0001 (D.W. and K.B.)), and the A.P. Moller Foundation. Y.-M.C. and S.B.S. are supported by NIH (1P50HD104224-01 and 2R37HD043341), the FDA (R01 FD007843), and the Gates Foundation (INV-024200). R.B. is supported by NIH (P50HD104224-01 and R01DE034452) and the Gates Foundation (INV-024200). C.M.L. is supported by the Li Ka Shing Foundation, NIHR Oxford Biomedical Research Centre, Oxford, NIH (1P50HD104224-01), Gates Foundation (INV-024200) and a Wellcome Trust Investigator Award (221782/Z/20/Z). This research was supported by the Wellcome Trust Core Award grant no. 203141/Z/16/Z with additional support from the NIHR Oxford BRC. The views expressed are those of the authors and not necessarily those of the NHS, the NIHR or the Department of Health.

## Author contributions

S.S.V. conceptualized the study, curated and analyzed data, supervised analyses and wrote the manuscript. L.B.L.W. conceptualized the study, curated and analyzed data and supervised analyses. D.S.P. and T.F. analyzed UKBB data and supervised analyses. N.A.B., B.H. and F.H.L. analyzed UKBB data. M.J.P. analyzed data. S.R. analyzed data and supervised analyses. P.D.R. analyzed Copenhagen Hospital Biobank and Danish Blood Donor Study (CHB/DBDS) data. M. Nyegaard supervised analyses of CHB/DBDS data. O.B.P., C.E., M.T.B., B.A.J., C.M., S.R.O., E.S., H.U., H.S.N., K.B., D.W. and A.J. provided resources from CHB/DBDS. B.M.J. and G.K. analyzed G&H data. D.A.v.H. supervised analyses and provided resources from G&H. J.F. analyzed EstBB data. R.M. supervised analyses and provided resources from EstBB. M.K.K. and A.P. analyzed FinnGen data. H.L. supervised analyses and provided resources from FinnGen. A.E. analyzed ALSPAC data. N.J.T., D.A.L. and A.F. supervised analyses and provided resources from ALSPAC. V.S. and G.T. analyzed deCODE data. H.H., I.J., I.O., A.H., T.S. and K.S. provided resources from deCODE. N.H. and D.I.B. provided resources. Y.-M.C., M.L. and R.B. contributed to defining phenotypes and provided clinical interpretation of results. S.B.S. provided clinical interpretation of results and acquired funding. T.L. supervised analyses, provided resources from EstBB, and wrote the manuscript. C.M.L. conceptualized the study and acquired funding. All authors read and approved the manuscript.

## Competing interests

L.B.L.W. is currently employed by Novo Nordisk Research Centre Oxford but, while she conducted the research described in this manuscript, was only affiliated to the University of Oxford. V.S., G.T., H.H., I.J. and K.S. are employees of deCODE genetics, a subsidiary of Amgen. C.M.L. is a part-time employee of Population Health Partners, owns equity in Population Health Partners and its subsidiaries, reports grants from Bayer AG and Novo Nordisk and has a partner who works at Vertex. The other authors declare no competing interests.

## Additional information

**Correspondence and requests for materials** should be addressed to Samvida S. Venkatesh or Cecilia M. Lindgren.

# Reporting Summary

## Statistics

For all statistical analyses, confirm that the following items are present in the figure legend, table legend, main text, or Methods section.

| n/a | Confirmed | |
|---|---|---|
| ☐ | ☒ | The exact sample size (*n*) for each experimental group/condition, given as a discrete number and unit of measurement |
| ☐ | ☒ | A statement on whether measurements were taken from distinct samples or whether the same sample was measured repeatedly |
| ☐ | ☒ | The statistical test(s) used AND whether they are one- or two-sided *Only common tests should be described solely by name; describe more complex techniques in the Methods section.* |
| ☐ | ☒ | A description of all covariates tested |
| ☐ | ☒ | A description of any assumptions or corrections, such as tests of normality and adjustment for multiple comparisons |
| ☐ | ☒ | A full description of the statistical parameters including central tendency (e.g. means) or other basic estimates (e.g. regression coefficient) AND variation (e.g. standard deviation) or associated estimates of uncertainty (e.g. confidence intervals) |
| ☐ | ☒ | For null hypothesis testing, the test statistic (e.g. *F*, *t*, *r*) with confidence intervals, effect sizes, degrees of freedom and *P* value noted *Give P values as exact values whenever suitable.* |
| ☐ | ☒ | For Bayesian analysis, information on the choice of priors and Markov chain Monte Carlo settings |
| ☒ | ☐ | For hierarchical and complex designs, identification of the appropriate level for tests and full reporting of outcomes |
| ☒ | ☐ | Estimates of effect sizes (e.g. Cohen's *d*, Pearson's *r*), indicating how they were calculated |

*Our web collection on statistics for biologists contains articles on many of the points above.*

## Software and code

Policy information about availability of computer code

| Data collection | No software was used specifically for data collection in this study; each biobank project that provided data for this study reported their data collection strategies in flagship papers that are cited in the Supplementary Methods section of this manuscript (Supplementary Information p30-33). |
|---|---|
| Data analysis | Each analysis described uses publicly available software as detailed in the relevant sub-sections of the Methods in the main manuscript and Supplemental Text.<br><br>Genome-wide association testing was performed independently be each cohort using REGENIE, SAIGE, or custom pipelines (versions differ by cohort and are reported in Supplementary Methods). GWASs in CHB/DBDS were performed with SAIGE v1.18. GWASs in EstBB were performed using the REGENIE software v3.0.3. GWASs in FinnGen was performed using the REGENIE software v2.2.4. GWASs in G&H were performed using the REGENIE software. GWASs in ALSPAC were performed using BOLT-LMM. GWASs in UK Biobank were performed using the REGENIE software v3.0.3.<br><br>GWAS meta-analysis was performed in METAL (version released 2011-03-25). Conditional analysis to identify novel SNPs was performed in GCTA-COJO v1.91.5. LD-score regression for heritability and genetic correlation analyses was performed using LDSC v1.0.0. Local genetic correlations were tested using LAVA v1.8.0. Polygenic overlap was assessed with MiXeR v1.3. Mendelian randomisation was performed using the TwoSampleMR v0.5.7 package in R. Colocalisation analyses were performed using the coloc v5.1.0 package in R. Publicly available single-cell RNA sequencing data was processed using the Seurat v3.0 package in R. Exome sequencing variant annotation was performed using VEP v105 with plugins for LOFTEE v1.04_GRCh38 and dbNSFP (annotating variants with CADD v1.6 and REVEL using dbNSFP4.3). SpliceAI v1.3 gene annotations were used to ensure alignment between VEP and SpliceAI. Rare variant genetic association testing using the exome sequeincng data was conducted in SAIGE version wzhou88/saige:1.1.9. The genotype data was LD pruned using PLINK v2.0. |

All other code to generate figures and tables was written in R v4.1.2.

Code Availability - All code used in this study is available through GitHub - https://github.com/lindgrengroup/infertility_hormones and Zenodo - https://doi.org/10.5281/zenodo.14528258.

For manuscripts utilizing custom algorithms or software that are central to the research but not yet described in published literature, software must be made available to editors and reviewers. We strongly encourage code deposition in a community repository (e.g. GitHub). See the Nature Portfolio guidelines for submitting code & software for further information.

## Data

Policy information about availability of data

All manuscripts must include a data availability statement. This statement should provide the following information, where applicable:

- Accession codes, unique identifiers, or web links for publicly available datasets
- A description of any restrictions on data availability
- For clinical datasets or third party data, please ensure that the statement adheres to our policy

Cohorts may be contacted individually for access to raw data. Meta-analysis summary statistics for all phenotypes are available through the GWAS Catalog study accession numbers GCST90483463 to GCST90483502.

## Research involving human participants, their data, or biological material

Policy information about studies with human participants or human data. See also policy information about sex, gender (identity/presentation), and sexual orientation and race, ethnicity and racism.

| | |
|---|---|
| Reporting on sex and gender | All analyses reported in this study are disaggregated by sex (i.e. male-specific and female-specific analyses only). Each cohort only retained individuals whose self-reported sex matched genetically-inferred sex from the genotyping or exome sequencing data (XX chromosomes for females and XY chromosomes for males). Sample sizes for each sex within each cohort are reported in Supp. Tables 2 (for infertility analyses) and 9 (for hormone analyses). Throughout the paper, we use the terms woman/female and man/male inter-changeably to refer to individuals' genetically inferred sex. We recognise the limitations of this binary analysis that excludes participants whose self-reported sex or gender did not match their genetically-inferred sex. |
| Reporting on race, ethnicity, or other socially relevant groupings | As genetic association testing must account for linkage disequilibrium (LD) patterns that differ based on ancestral population histories, we conducted analyses separately in different population groups. Each biobank project that provided data for this study assigned genetically-inferred population labels (referred to in this paper as "ancestry") to samples based on different strategies reported in their flagship papers that are cited in the Methods section of this manuscript (Supp. Text p5-6). Sample sizes for each ancestry group are specified in Supp. Tables 2 (for infertility analyses) and 9 (for hormone analyses). For the UK Biobank, we assigned sample population labels by training a random forest (RF) classifier using the 1000 Genomes 'super-population' labels. We first ran principal components analysis (PCA) on unrelated individuals in the 1000 Genomes project dataset, subset to LD-pruned autosomal variants. Samples in the UKBB genotyping data are projected onto this PCA space, ensuring that we correctly account for shrinkage bias in the projection. Next, we used the 'super-population' labels (AFR=Africans, AMR=Admixed Americans, EAS=East Asians, Europeans=EUR, South Asians=SAS) of the 1000 Genomes dataset to train a RF classifier, using the randomForest (4.6) library in R, and predicted the super-population for each of the UKBB samples. Samples with classification probability >0.99 were retained for downstream analysis. Genome-wide association studies were performed separately in each genetically-inferred ancestral group. Meta-analyses were performed in two sets: (1) EUR-only subset, (2) all-ancestries. Any analyses that are dependent on LD are performed in the EUR-only subset and are flagged as such (Figure 1). |
| Population characteristics | Population characteristics (sample size, genotyping information, and types of records used for case/control ascertainment) for each biobank project included in this study are described in Supplementary Information pages 30-33. Sex- and ancestry-specific sample sizes are reported in Supp. Tables 2 and 9. |
| Recruitment | Each biobanking project used in this study followed different recruitment protocols, which are detailed in their flagship papers. A brief summary is provided in the Supplementary Information pages 30-33 along with citations to the flagship papers of each cohort. |
| Ethics oversight | This study was conducted using data from seven research studies, each with separate ethics approval. 1. UK Biobank has approval from the North West Multi-centre Research Ethics Committee (MREC) as a Research Tissue Bank (RTB) approval. 2. Copenhagen Hospital Biobank is classified as a biobank for future research. It is part of the Danish National Biobank and has been approved by the Danish Data Protection Agency (general approval number 2012-58-0004, and local number: RH-2007-30-4129/I-suite 00678). The Danish Blood Donor Study was approved by the Central Denmark (1-10-72-95-13) and Zealand (SJ-740) Regional Committees on Health Research Ethics and the Data Protection Agency (P-2019-99). The DBDS GWA study was approved by the Danish National Committee on Health Research Ethics (1700407). SCANDAT was approved by the Data Protection Agency (2008-54-0472). The DBDS was conducted in accordance with the ethical principles outlined in the Declaration of Helsinki. 3. Genes & Health operates under ethical approval, 14/LO/1240, from London South East NRES Committee of the Health Research Authority, dated 16 September 2014. 4. The activities of the Estonian Biobank are regulated by the Human Genes Research Act, which was adopted in 2000 specifically for the operations of the EstBB. Individual level data analysis in the EstBB was carried out under ethical approval 1.1-12/624 from the Estonian Committee on Bioethics and Human Research (Estonian Ministry of Social Affairs), using data according to release application 3-10/ GI/10790 from the Estonian Biobank. 5. Patients and control subjects in FinnGen provided informed consent for biobank research, based on the Finnish Biobank Act. Alternatively, separate research cohorts, collected prior the Finnish Biobank Act |

came into effect (in September 2013) and start of FinnGen (August 2017), were collected based on study-specific consents and later transferred to the Finnish biobanks after approval by Fimea (Finnish Medicines Agency), the National Supervisory Authority for Welfare and Health. Recruitment protocols followed the biobank protocols approved by Fimea. The Coordinating Ethics Committee of the Hospital District of Helsinki and Uusimaa (HUS) statement number for the FinnGen study is Nr HUS/990/2017. The FinnGen study is approved by Finnish Institute for Health and Welfare (permit numbers: THL/2031/6.02.00/2017, THL/1101/5.05.00/2017, THL/341/6.02.00/2018, THL/2222/6.02.00/2018, THL/283/6.02.00/2019, THL/1721/5.05.00/2019 and THL/1524/5.05.00/2020), Digital and population data service agency (permit numbers: VRK43431/2017-3, VRK/6909/2018-3, VRK/4415/2019-3), the Social Insurance Institution (permit numbers: KELA 58/522/2017, KELA 131/522/2018, KELA 70/522/2019, KELA 98/522/2019, KELA 134/522/2019, KELA 138/522/2019, KELA 2/522/2020, KELA 16/522/2020), Findata permit numbers THL/2364/14.02/2020, THL/4055/14.06.00/2020, THL/3433/14.06.00/2020, THL/4432/14.06/2020, THL/5189/14.06/2020, THL/5894/14.06.00/2020, THL/6619/14.06.00/2020, THL/209/14.06.00/2021, THL/688/14.06.00/2021, THL/1284/14.06.00/2021, THL/1965/14.06.00/2021, THL/5546/14.02.00/2020, THL/2658/14.06.00/2021, THL/4235/14.06.00/2021, Statistics Finland (permit numbers: TK-53-1041-17 and TK/143/07.03.00/2020 (earlier TK-53-90-20) TK/1735/07.03.00/2021, TK/3112/07.03.00/2021) and Finnish Registry for Kidney Diseases permission/extract from the meeting minutes on 4th July 2019. 6. Ethical approval for the study was obtained from the Avon Longitudinal Study of Parents and Children (ALSPAC) Ethics and Law Committee and the Local Research Ethics Committees. 7. The deCode Genetics study was approved by the Icelandic National Bioethics Committee (approval no. VSN-19-023).

Note that full information on the approval of the study protocol must also be provided in the manuscript.

# Field-specific reporting

Please select the one below that is the best fit for your research. If you are not sure, read the appropriate sections before making your selection.

☒ Life sciences ☐ Behavioural & social sciences ☐ Ecological, evolutionary & environmental sciences

For a reference copy of the document with all sections, see nature.com/documents/nr-reporting-summary-flat.pdf

# Life sciences study design

All studies must disclose on these points even when the disclosure is negative.

| | |
|---|---|
| Sample size | Sample sizes for infertility GWAS meta-analyses: up to 42,629 cases / 740,619 controls in women, and 10,886 cases / 995,982 controls in men; sample sizes for hormone GWAS meta-analyses: up to 246,862 for testosterone in women, and 243,951 for testosterone in men. Detailed sample sizes for all subtypes of infertility and hormones are outlined in Supp. Tables 2 & 9. Sample sizes were not pre-determined as there was no minimum effect size we were trying to capture. We were looking to identify the smallest significant effect size possible with the available samples. Each of the seven study cohorts provided the maximum number of samples possible for each phenotype. |
| Data exclusions | Each cohort performed variant- and sample-specific quality control before genome-wide association testing; these are described in Supp. Text p8-10. Prior to meta-analysis, we performed additional variant-specific QC to retain variants that met the following criteria: (1) on the autosomes or X chromosome, (2) with imputation information score >0.8, (3) bi-allelic variants with A,C,G,T alleles, (4) with standard errors <10 and P-values in [0,1], and (5) without duplicate entries. Variants and samples for exome sequencing analyses underwent separate quality control, as outlined on p32 of the main text and p11 of Supp. Text. Briefly, we first considered an initial set of "high quality" variants to evaluate the mean call rate and depth of coverage for each sample. We then ran a sample and variant level pre-filtering step and calculated sample-level QC metrics. Using these metrics, we removed sample outliers based on median absolute deviation (MAD) thresholds, and excluded sites which did not pass variant QC according to Karczewski et al. (2022). We then applied a genotype-level filter using genotype quality (GQ), depth (DP), and heterozygote allele balance (AB). The resultant high-quality European call set consisted of 402,375 samples and 25,229,669 variants. |
| Replication | As this study reports a meta-analysis of GWASs from several biobanking projects, we did not seek additional replication. For all lead variants reported in the manuscript, we report the heterogeneity in effect sizes among participating cohorts; we also flag if a lead variant was only reported in one cohort. We sought replication of the exome-sequencing results (discovery in UK Biobank) at the variant level and gene level. At the variant level, we looked up the effects of rare variants in EUR-ancestry GWAS meta-analysis excluding UKB participants (hence, replicated once). At the gene level, we report the results from gene-burden testing in the deCODE and Genes & Health datasets (hence, replicated twice). Supp. Tables 15, 20, and 21 include replication results.<br><br>In discovery, gene-burden analyses implicated the PLEKHG4 gene for F-EXCL (burden test OR (95% CI)=1.04 (1.02-1.06) when aggregated across all variant annotations with MAF<1%, Cauchy P=2.47E-07). This association did not replicate in the deCODE or Genes & Health (G&H) datasets (P>0.05).<br><br>We discovered 43 gene-trait pairs associated with testosterone with Cauchy P<5E-06 in UK Biobank, thirteen (30.2%) of which replicate at nominal significance (P<0.05), and two (4.65%) at Bonferroni-adjusted significance (P<6.85E-04) in either the deCODE or G&H datasets with consistent directions of effect. |
| Randomization | The current work was not an experimental study so did not require randomisation into experimental groups. |
| Blinding | The current work was not an experimental study so did not require blinding to group allocation during data collection or analysis. |

# Reporting for specific materials, systems and methods

We require information from authors about some types of materials, experimental systems and methods used in many studies. Here, indicate whether each material, system or method listed is relevant to your study. If you are not sure if a list item applies to your research, read the appropriate section before selecting a response.

## Materials & experimental systems

| n/a | Involved in the study |
|-----|----------------------|
| ☒ ☐ | Antibodies |
| ☒ ☐ | Eukaryotic cell lines |
| ☒ ☐ | Palaeontology and archaeology |
| ☒ ☐ | Animals and other organisms |
| ☒ ☐ | Clinical data |
| ☒ ☐ | Dual use research of concern |
| ☒ ☐ | Plants |

## Methods

| n/a | Involved in the study |
|-----|----------------------|
| ☒ ☐ | ChIP-seq |
| ☒ ☐ | Flow cytometry |
| ☒ ☐ | MRI-based neuroimaging |

## Plants

| | |
|---|---|
| Seed stocks | *Report on the source of all seed stocks or other plant material used. If applicable, state the seed stock centre and catalogue number. If plant specimens were collected from the field, describe the collection location, date and sampling procedures.* |
| Novel plant genotypes | *Describe the methods by which all novel plant genotypes were produced. This includes those generated by transgenic approaches, gene editing, chemical/radiation-based mutagenesis and hybridization. For transgenic lines, describe the transformation method, the number of independent lines analyzed and the generation upon which experiments were performed. For gene-edited lines, describe the editor used, the endogenous sequence targeted for editing, the targeting guide RNA sequence (if applicable) and how the editor was applied.* |
| Authentication | *Describe any authentication procedures for each seed stock used or novel genotype generated. Describe any experiments used to assess the effect of a mutation and, where applicable, how potential secondary effects (e.g. second site T-DNA insertions, mosiacism, off-target gene editing) were examined.* |

