## [Peer Review File · Nature Genetics]

Genome-wide analyses identify 25 infertility loci and relationships with reproductive traits across the allele frequency spectrum

Corresponding Author: Dr Samvida Venkatesh

Version 0:

Decision Letter:

31st May 2024

Dear Dr. Venkatesh,

Your Article "Genome-wide analyses identify 21 infertility loci and over 400 reproductive hormone loci across the allele frequency spectrum" has been seen by two referees. You will see from their comments below that, while they find your work of interest, they have raised several relevant points. We are interested in the possibility of publishing your study in Nature Genetics, but we would like to consider your response to these points in the form of a revised manuscript before we make a final decision on publication.

To guide the scope of the revisions, the editors discuss the referee reports in detail within the team, including with the chief editor, with a view to identifying key priorities that should be addressed in revision, and sometimes overruling referee requests that are deemed beyond the scope of the current study. In this case, we ask that you address all technical queries related to the association analyses and their interpretation, in particular exploring possible ascertainment biases in hormone level measurements and their potential impact on downstream analyses as requested by Reviewer #1 and applying LAVA and MiXeR to explore local genetic correlations as requested by Reviewer #2. We hope you will find this prioritized set of referee points to be useful when revising your study. Please do not hesitate to get in touch if you would like to discuss these issues further.

We therefore invite you to revise your manuscript taking into account all reviewer and editor comments. Please highlight all changes in the manuscript text file. At this stage, we will need you to upload a copy of the manuscript in MS Word .docx or similar editable format.

*2) If you have not done so already, please begin to revise your manuscript so that it conforms to our Article format instructions, available

[here](http://www.nature.com/ng/authors/article_types/index.html).

*3) Include a revised version of any required Reporting Summary: <https://www.nature.com/documents/hr-reporting-summary.pdf>

Please be aware of our [guidelines on](https://www.nature.com/nature-research/editorial-policies/image-integrity)

digital image standards.

Link Redacted

We hope to receive your revised manuscript within 8-12 weeks. If you cannot send it within this time, please let us know.

Nature Genetics is committed to improving transparency in authorship. As part of our efforts in this direction, we are now requesting that all authors identified as 'corresponding author' on published papers create and link their Open Researcher and Contributor Identifier (ORCID) with their account on the Manuscript Tracking System (MTS), prior to acceptance. ORCID helps the scientific community achieve unambiguous attribution of all scholarly contributions. You can create and link your ORCID from the home page of the MTS by clicking on 'Modify my Springer Nature account'. For more information, please visit www.springernature.com/orcid.

Sincerely,
Kyle

Kyle Vogan, PhD
Senior Editor
Nature Genetics
<https://orcid.org/0000-0001-9565-9665>

Referee expertise:

Referee #1: Genetics, reproductive traits, statistical methods

Referee #2: Genetics, reproductive traits,

Reviewers' Comments:

Reviewer #1:
Remarks to the Author:

The paper describes the largest GWAS meta-analysis to date of infertility in both males and females, identifying 21 genetic loci of which 12 are novel. The strength of this paper is the contribution to advancing understanding of the genetics of infertility, made possible by deriving phenotypes from electronic health records (EHR) data across 6 cohorts. Considerable further analyses test correlations and causal relationships with other phenotypes, investigate the evolutionary basis of the infertility signals and explore the role of rare coding variation. As part of these additional analyses, GWAS of sex hormones derived from EHRs and direct measures are performed across multiple studies, expanding the sample size for these phenotypes – however I have some concerns about the approach (see below). Several additional genes are implicated by rare variant analyses of hormones and infertility. Interestingly, the authors find little general overlap in the genetic basis of sex hormone regulation and infertility, an interesting finding.

Code has not been tested (not yet available) but methods seem appropriate and described in sufficient detail.

Main comments:

- 1) For the lead signals, are similar effects seen across phenotypes and do any signals show nominal association in other phenotypes? This may be informative. Are results consistent between the inclusion and exclusion definition for female idiopathic infertility?
- 2) Given that age at menopause is considered a proxy of ovarian reserve, do any of the genetic signals overlap with those for age at menopause? Figure 3B seems to suggest a genetic correlation between AMH and anovulatory infertility, though this is unclear as line 350 states $P > 0.05$. AMH has been shown to have a strong genetic correlation with menopause timing (Verdiesen et al. 2022) – is the AMH/infertility correlation driven by age at menopause?
- 3) To what extent are the GWAS signals for FSH, LH, etc. influenced by biases inherent in the health records data such as

underlying medical conditions, and reasons for carrying out hormone tests? For example, DCLRE1A and CHEK2 identified for FSH have been implicated in menopause timing (e.g. Ward, L. D. et al. HGG Adv 3, (2021)) and FSH is measured clinically in the diagnosis of premature menopause. Are the genetic signals for the hormone phenotypes actually capturing other phenotypes? This may affect downstream analyses.

4) In Supp Table 13 (Genetic Correlations), the phenotype "is female" appears to be correlated with male testosterone. Can the authors comment as to why this would be the case?

5) For the MR analyses, it would be helpful to include Egger intercept results. Also, for FSH where small numbers of SNPs are analysed, it would be helpful to see scatter plots of effect size plus CI for the SNPs in the MR analyses with the fitted estimates.

6) Ideally the exome analyses results should be replicated. Is there any overlap in allele frequency in the variants included in the exome analyses and GWAS and do any of the exome associations replicate in any of the other cohorts?

Minor comments:

The abbreviation of GWS for genome-wide signal seems unnecessary

Supp Table 1: it would be helpful to have text descriptions of what conditions/diseases were considered to fall into the different categories of infertility.

Supp Table 2 and Supp Table 9 – please add to the paper the total sample size for the meta-analysis of each phenotype.

Supp Table 3, Supp Table 10, Supp Table 15 – please add the effect allele and clarify which is the effect allele tested and the frequency of that allele. Likewise, Table 1 (i.e. is A1 the minor allele?).

How were the genes mapped to the GWAS signals in Table 1?

Please add effect allele and frequency to discussion of variants in the text (e.g. to line 184-185).

Did the genome-wide results show any evidence of genomic inflation? Useful to include LDSC intercept in Supp Table 6.

Line 307-310 – Why were other reproductive associations not considered?

Line 349 – Figure 3 legend suggests that AMH is at $P < 0.05$

Tables and Supplementary Tables: it would be helpful to state the Bonferroni corrected p-thresholds in the headers.

Supp Table 13 it would be helpful to provide results for all 703 traits tested.

Line 390 – I couldn't match the Cauchy F-EXCL p-value to Supp Table 14.

Figure 6A – it was confusing that the points are sized by beta as well as having this on the y axis

Supplementary Table 15 – it would be helpful to add the ID and MAF of the common variant

Ref 116 – issue with formatting

Supplementary Methods: This line does not make sense "GWASs in the UKBB were additionally adjusted for one-hot encoding of data provider or assessment centre"

Reviewer #2:

Remarks to the Author:

I commend the authors for their novel, comprehensive, and very well-written study.

I have no major concerns or issues with the presented research and provide the following comments in line with the *Nature Genetics* review process and goal to improve and/or extend the interpretation of the study findings.

Summary of the key results

The key results are both appropriate and reasonable in their emphasis. The manuscript most certainly provides "the first comprehensive view of the genetic architecture of infertility".

However, I question the conclusion that their "results suggest that while individual genes associated with hormone regulation may be relevant for fertility, there is limited genetic evidence for correlation between reproductive hormones and infertility at the population level" (Page 3, Line 105-106). See 'Suggested improvements'.

Originality and significance: if not novel, please include reference

The research and findings are highly novel. In addition to the identified novel risk loci and genes, the GWAS summary statistics resulting from this study (made available through the GWAS Catalog upon publication) will prove extremely valuable for future research aimed at understanding the role and relationship of infertility and reproductive hormones in other human traits.

Data & methodology: validity of approach, quality of data, quality of presentation

The study utilised current and appropriate methodology, the data was of high quality and presented in a very clear and concise manner.

Appropriate use of statistics and treatment of uncertainties

The employed statistical tests were appropriate and correctly interpreted. The study was very well-powered and does not overstate its findings.

Conclusions: robustness, validity, reliability

The results are carefully interpreted and discussed with respect to current knowledge. The conclusion are appropriate, with perhaps the exception regarding "limited genetic evidence for correlation between reproductive hormones and infertility at the population level" (see 'Suggested improvements').

Suggested improvements: experiments, data for possible revision

Major comment:

The authors make numerous important statements regarding (absence of) correlation between traits at the population level that are based largely, if not solely, on LDSC genetic correlation (r_g) analyses. However, given LDSC assesses the global (genome-wide) r_g , it only considers and quantifies the average sharing of genetic effects between a pair of traits. Therefore, when r_g is confined to a particular genomic region(s) or is in opposite directions at varying loci, LDSC may not detect a significant r_g .

Indeed, the authors state that "Only 9.80% (of 153 total) lead variants for testosterone in females and 5.75% (of 261 total) lead variants for testosterone in males reach GWS in both sexes; and 45.9% of variants have opposing directions of effect in men and women (Supp. Figure 6). Indeed, we found no significant genetic correlation between testosterone in men and women (r_g (SE)=0.0361 (0.0428), $P=0.399$)." (Page 13, Line 332-335). However, I would argue that 9.8% and 5.75% is a non-negligible (and significant) overlap of genome-wide significant risk loci. Furthermore, the observation that "45.9% of variants have opposing directions of effect in men and women" would perfectly explain why the r_g between testosterone in men and women was not significant (i.e., the averaging of positive and negative effects result in a non-significant r_g).

Similarly, an over-reliance on LDSC r_g in determining a (polygenic) relationship between traits extends to the "limited genetic evidence for correlation between reproductive hormones and infertility" (Page 3, Line 105-106); "24 obesity-related traits, including body mass index (BMI), waist-to-hip ratio (WHR), and body fat percentage, are correlated with testosterone and FSH, but are not genetically correlated with any category of female infertility" (Page 14, Line 367-369); "we did not find substantial genetic correlation between obesity and infertility" (Page 17, Line 477-478); and "We did not find evidence that reproductive hormone dysregulation and obesity are strongly correlated with infertility at the population level, but instead nominate individual hormone-associated genes with effects on fertility" (Page 18, Line 525-527).

Thus, use of recently developed approaches such as LAVA (Local Analysis of [co]Variant Association) [Werme et al 2022 doi:10.1038/s41588-022-01017-y] could help examine local r_g . This will allow the examination of and provide insight into their complex genetics.

Also, the 'MiXeR' method quantifies polygenic overlap, using GWAS summary statistics, irrespective of (global) genetic correlation (Frei et al 2019 doi: 10.1038/s41467-019-10310-0).

Indeed, I suggest (argue) that in addition to "Other genetic and non-genetic avenues must be explored to treat complex and heterogeneous fertility disorders that impact the physical, emotional, and financial well-being of millions of individuals across the globe" (Page 19, Line 527-529) these additional genetic analyses (LAVA and MiXeR) should be performed for the current study.

Minor comments:

It was not clear how the lead "SNPs were annotated to the mapped gene" - can the authors please add these details.

Re: the statement "Finally, we identified 29 unique genes carrying rare variants ($MAF < 1\%$) associated with testosterone in male or female participants in the UK Biobank. Eighteen of the 29 genes also contain common testosterone-associated variants from GWASs ($MAF > 1\%$), but the rare variant has a larger absolute effect size in the majority (83%) of these" (Page 15, Line 413-416): could the authors please comment on whether the common and rare variants at the same loci are statistical independent and/or are perhaps the result of synthetic associations (e.g., see Wray et al 2011 doi: 10.1371/journal.pbio.1000579)?

The statement "The strong genetic correlation of 71% between idiopathic infertility and endometriosis may indicate that

some proportion of idiopathic cases are due to under-diagnosis of endometriosis, whose early treatment may prevent future infertility" (line 467-469)

References: appropriate credit to previous work?

The references appear appropriate and sufficient.

Clarity and context: lucidity of abstract/summary, appropriateness of abstract, introduction and conclusions

All aspects of the manuscript are very clear and contextualised.

Version 1:

Decision Letter:

Our ref: NG-A65009R

3rd October 2024

Dear Samvida,

Your revised manuscript "Genome-wide analyses identify 25 infertility loci and relationships with reproductive traits across the allele frequency spectrum" (NG-A65009R) has been seen by the original referees. As you will see from their comments below, the reviewers are satisfied with the revision, and therefore we will be happy in principle to publish your study in Nature Genetics as an Article pending final revisions to comply with our editorial and formatting guidelines.

We are now performing detailed checks on your paper, and we will send you a checklist detailing our editorial and formatting requirements soon. Please do not upload the final materials or make any revisions until you receive this additional information from us.

Thank you again for your interest in Nature Genetics. Please do not hesitate to contact me if you have any questions.

Sincerely,
Kyle

Kyle Vogan, PhD
Senior Editor
Nature Genetics
<https://orcid.org/0000-0001-9565-9665>

Reviewer #1 (Remarks to the Author):

This remains an interesting and important paper that will make a valuable addition to the field. I would like to thank the authors for their thorough and clear responses and extensive additional analyses.

One minor comment:

Line 424: I suggest re-wording slightly – as I understand it, in the UK FSH is still used to diagnose premature menopause but is not recommended for diagnosis of menopause in the normal age range. See <https://cks.nice.org.uk/topics/menopause/diagnosis/diagnosis-of-menopause-perimenopause/>

Otherwise I have no further comments.

Reviewer #2 (Remarks to the Author):

I thank the authors for their careful and detailed responses to my comments.

The authors have performed additional analyses (as per my suggestions/requests) and appropriately amended their manuscript to my satisfaction.

The resulting revised manuscript is very balanced and scientifically robust.

I have no further comments, other than to say: "well done, nice work!".

Editor Comments:

To guide the scope of the revisions, the editors discuss the referee reports in detail within the team, including with the chief editor, with a view to identifying key priorities that should be addressed in revision, and sometimes overruling referee requests that are deemed beyond the scope of the current study. In this case, we ask that you address all technical queries related to the association analyses and their interpretation, in particular exploring possible ascertainment biases in hormone level measurements and their potential impact on downstream analyses as requested by Reviewer #1 and applying LAVA and MiXeR to explore local genetic correlations as requested by Reviewer #2. We hope you will find this prioritized set of referee points to be useful when revising your study. Please do not hesitate to get in touch if you would like to discuss these issues further.

Thank you very much for your careful consideration of our work. We have provided a point-by-point response to both reviewers below, taking particular care to address concerns around ascertainment bias in hormone measurements and incorporate results from local genetic correlation analyses.

Reviewer #1:**Remarks to the Author:**

The paper describes the largest GWAS meta-analysis to date of infertility in both males and females, identifying 21 genetic loci of which 12 are novel. The strength of this paper is the contribution to advancing understanding of the genetics of infertility, made possible by deriving phenotypes from electronic health records (EHR) data across 6 cohorts. Considerable further analyses test correlations and causal relationships with other phenotypes, investigate the evolutionary basis of the infertility signals and explore the role of rare coding variation. As part of these additional analyses, GWAS of sex hormones derived from EHRs and direct measures are performed across multiple studies, expanding the sample size for these phenotypes – however I have some concerns about the approach (see below). Several additional genes are implicated by rare variant analyses of hormones and infertility. Interestingly, the authors find little general overlap in the genetic basis of sex hormone regulation and infertility, an interesting finding.

Code has not been tested (not yet available) but methods seem appropriate and described in sufficient detail.

Thank you for taking the time to generously review our work. We have now made the code on GitHub publicly available here - https://github.com/lindgrengroup/infertility_hormones and revised the “Code availability” statement to reflect this (**Lines 1060-1061**).

Main comments:

1) For the lead signals, are similar effects seen across phenotypes and do any signals show nominal association in other phenotypes? This may be informative. Are results consistent between the inclusion and exclusion definition for female idiopathic infertility?

Thank you for this valuable suggestion. We have made the following revisions to incorporate the result on heterogeneity among lead SNP effects across different infertility categories.

Revised text:

Lines 185-187: “Fourteen loci (63.6%) for female infertility reached nominal significance ($P < 2.27E-03$, Bonferroni correction for 22 independent loci tested) in at least one other infertility category (Supp. Text, Supp. Figure 19).”

Supplementary Text (p1, para 1): “We assessed the consistency of lead SNP effects between the two definitions of idiopathic infertility, i.e. by excluding known causes such as anovulation, anatomical, PCOS, endometriosis, and uterine fibroids (F-EXCL) and by inclusion of a diagnostic code for idiopathic infertility (F-INCL). Two lead SNPs were shared between these definitions (i.e. rs61768001 mapped to *WNT4* and rs17378154 mapped to *PKHD1L1/EBAG9*, which are discussed in the Main text. Another two lead SNPs associated with F-INCL reach nominal significance ($P < 2.27E-03$, Bonferroni adjustment for 22 unique lead SNPs tested across all categories) for F-EXCL, i.e. rs74156208 mapped to *TMEM26* and rs192462512 mapped to *BCOR*. All lead SNPs for F-EXCL and F-INCL are directionally consistent across both categories (Supp. Figure 19).”

Supp. Figure 19: Forest plots displaying the effect of lead SNPs associated with female infertility phenotypes across all categories of female infertility.

2) Given that age at menopause is considered a proxy of ovarian reserve, do any of the genetic signals overlap with those for age at menopause? Figure 3B seems to suggest a genetic correlation between AMH and anovulatory infertility, though this is unclear as line 350 states $P > 0.05$. AMH has been shown to have a strong genetic correlation with menopause timing (Verdiesen et al. 2022) – is the AMH/infertility correlation driven by age at menopause?

We thank the reviewer for this excellent suggestion. We performed colocalisation to test the genetic overlap among AMH, anovulatory infertility, and age at menopause, and found colocalisation between the latter two traits at the *INHBB* locus; inhibin B is itself a marker of ovarian reserve. As there was no significant genetic correlation (after multiple-testing) among the three traits, we have added the following to the Supplementary Text. We have also corrected the error on line 350 (now line 444) to state the nominal genetic correlation between AMH and F-ANOV.

Revised text:

Lines 439-440: “(all $P > 0.05$, except the correlation between AMH and F-ANOV, r_g (SE)=0.748 (0.301), $P=0.0131$; Figure 3B)”.

Supp. Text (p2, para 1): “As age at menopause is considered a proxy of ovarian reserve, and AMH is associated with menopause timing⁸², we assessed the genetic relationships between age at menopause, female infertility, and AMH. There was nominally significant ($P < 0.05$) positive genetic correlation between F-ANOV and AMH (r_G (SE)=0.748 (0.301), $P=0.0131$) and between F-ANOV and age at menopause (r_G (SE)=0.164 (0.0789), $P=0.0382$). We also tested for colocalisation¹³⁴ of genetic signals among these three traits and identified statistically significant overlap (posterior probability (PP) of shared causal

SNP=100%) between F-ANOV and age at menopause at the *INHBB* locus. However, this locus is not shared with AMH (PP=0.363%), as there is no evidence for an AMH association at the *INHBB* locus (minimum $P=0.119$). The association between increased risk of anovulatory infertility and older age at menopause at this locus is likely driven by inhibin B regulation, itself a marker of ovarian reserve.”

Supp. Table 5: Included colocalisation results between all categories of female infertility and age at menopause.

3) To what extent are the GWAS signals for FSH, LH, etc. influenced by biases inherent in the health records data such as underlying medical conditions, and reasons for carrying out hormone tests? For example, DCLRE1A and CHEK2 identified for FSH have been implicated in menopause timing (e.g. Ward, L. D. et al. HGG Adv 3, (2021)) and FSH is measured clinically in the diagnosis of premature menopause. Are the genetic signals for the hormone phenotypes actually capturing other phenotypes? This may affect downstream analyses.

We thank the reviewer for this comment. Indeed most of the cohorts in our analysis contributed measurements for FSH and LH that may have been ascertained, i.e. the hormones may have been measured to diagnose underlying medical conditions such as early age at menopause. To address this concern, we tested for the difference in age at menopause between women with and without clinical measurements of FSH or LH in the UK Biobank. We also checked whether GWAS signals may have been biased by testing for heterogeneity in the effects of the meta-analysis lead variants for FSH and LH on: (1) all 20,749 women in UKBB with these hormones measured, and (2) only the subset of 5,949 women with self-reported age at menopause >45 years (i.e., who did not experience “early” menopause).

Revised text:

Lines 400-412: “Although no longer recommended, clinical measurements of FSH and LH were used in the past to diagnose premature menopause,⁷⁹ so our hormone GWASs based on these measurements may have captured signals for menopause timing (12 lead variants for FSH and five for LH have previously been reported for associations with age at menopause). We found a significant (two-tailed t -test $P<0.05$), but small (0.2 years) difference between self-reported age at menopause in women with absence of clinical measurements of FSH or LH (mean (SE) age at menopause = 49.7 (0.0135) years) and those with measurements of FSH (mean=49.5 (0.0583) years, $P=0.0141$) and LH (49.5 (0.0695) years, $P=0.00155$) (Supp. Figure 18). This potential ascertainment bias did not appear to affect GWAS results – all lead SNPs for FSH and LH had consistent effects in the subset of 5,949 women in UKBB known to have undergone menopause at age >45 years (that is, they did not experience early menopause) as in the full dataset of 20,749 women without filtering on menopause status (heterogeneity $P>3.85E-03$, Bonferroni adjustment for 13 SNP-trait pairs tested, Supp. Table 25, Supp. Text).”

Supp. Text (p2, para 2): “As detailed in the Main text, clinical measurements of the gonadotropins FSH and LH may have been used to diagnose premature menopause. The lead SNPs for these hormones from our reported meta-analysis did not differ in their effect

sizes between GWASs based solely on the 5,949 women in UKBB with age at menopause >45 years (i.e. women who did not experience early menopause) as compared to GWASs in all 20,749 women in UKBB with measurements of these hormones (heterogeneity $P > 3.85E-03$, Bonferroni adjustment for 13 SNP-trait pairs tested, Supp. Table 25). However, as we may have missed sub-genome wide significant signals for these hormones, we also assessed the heterogeneity in variant effects among SNPs that reached $P < 1E-06$ for either the full set of women in UKBB or the subset that did not undergo early menopause (SNPs were distance-pruned, resulting in 109 independent SNP-trait pairs tested). Four SNPs show significant differences ($P < 4.59E-04$, Bonferroni adjustment for 109 tests) in effects between these two samples, all of which only reach significance in the subset of women who did not go through early menopause, and have between 5.98-11.0x larger effects in these women than the full dataset (Supp. Table 25). rs78114522 (MAF=1.77%, intron in *ZNF148*) is associated with FSH levels in women with menopause age >45 years (beta (SE)=-0.501 (0.104), $P=1.43E-06$); *ZNF148* is a transcriptional repressor expressed in the thyroid gland. Three variants are only associated with LH in the subset of women who undergo menopause at age >45 years: rs140409 (MAF=47.5%, beta (SE)=0.108 (0.0224), $P=1.60E-06$) is near *GGTLC2*, whose deficiency is associated with altered circulating FSH, impaired folliculogenesis, absent corpus luteum and tertiary ovarian follicles, and oocyte degeneration in mice¹⁵⁸ and with elevated thyroid levels in humans²⁷; rs113176873 (MAF=1.87%, beta (SE)=0.362 (0.079), $P=4.53E-06$) is near *PTK2B*, whose deletion improves folliculogenesis and fecundity, mediated by the gonadotropin-stimulating ERK cascade, in mouse models¹⁵⁹; and rs61412562 (MAF=36.1%, beta (SE)=-0.0966 (0.0217), $P=8.13E-06$) an intronic variant in *ZNF234*. While these are promising candidate genes to understand the relationships between gonadotropins and female infertility, none of the variants noted reach genome-wide significance in our analyses and must be replicated.”

Supp. Figure 18: Distribution of self-reported age at menopause in female participants in UK Biobank with or without measurements of follicle stimulating hormone (FSH) and luteinising hormone (LH) in the linked general practice (GP) data.

Supp. Table 25: Heterogeneity in the effects of SNPs associated with follicle stimulating hormone (FSH) and luteinising hormone (LH) between all women in UK Biobank and women with age at menopause > 45 years only.

4) In Supp Table 13 (Genetic Correlations), the phenotype “is female” appears to be correlated with male testosterone. Can the authors comment as to why this would be the case?

We thank the reviewer for bringing this to our attention. As reported in Supp. Table 13, the correlation between sex (“is_female”) and testosterone is r_G (SE)=0.233 (0.0503), $P=3.53E-06$. We believe this is a spurious correlation arising from sex-differential participation bias in UK Biobank.

Pirastu *et al.* (2021) report a GWAS of sex in UK Biobank to identify genetic contributors to sex-differential participation bias that leads to differences in autosomal allele frequencies between males and females in the study. In Supp. Table 6, they report that sex is genetically correlated with: height, birth weight, body mass index (BMI), LDL cholesterol, type 2

diabetes, breast cancer, educational attainment, major depressive disorder, risk-taking behaviour, and schizophrenia in UK Biobank (all $P < 0.05$).

For several of these traits, the authors show that genetically predicted trait value can lead to sex-differential participation bias. For example, they note that there is BMI-related participation bias in the UK Biobank; BMI-increasing autosomal alleles are more frequent in men than in women. As testosterone is correlated with BMI, it is likely that sex-differential study participation resulted in a spurious genetic correlation between testosterone and sex.

Revised text:

To prevent confusion, we have removed “is_female” as a trait in Supp. Table 13.

5) For the MR analyses, it would be helpful to include Egger intercept results. Also, for FSH where small numbers of SNPs are analysed, it would be helpful to see scatter plots of effect size plus CI for the SNPs in the MR analyses with the fitted estimates.

Thank you for these suggestions. We have added Egger intercepts and their P -values to Supplementary Table 11, and scatter plots for the effect sizes and CIs of SNPs included in MR analyses for FSH on infertility (Supp. Figure 15).

Revised text:

Lines 442-444: “Although these appear to be driven by a single outlier variant (rs150206026), leave-one-out sensitivity analyses indicate that the MR effect remains consistent upon excluding this variant (MR $P < 0.00158$) (Supp. Figure 15).”

Supp. Table 11: Now includes MR-Egger intercept and P -value.

Supp. Figure 15: Scatter plots displaying genetic variants included in the female follicle stimulating hormone (FSH) instrument for Mendelian Randomisation (MR) analyses.

6) Ideally the exome analyses results should be replicated. Is there any overlap in allele frequency in the variants included in the exome analyses and GWAS and do any of the exome associations replicate in any of the other cohorts?

Thank you for the request to replicate our analyses, which offered us the ability to greatly strengthen our results. We replicated the exome analyses at:

1. the variant-level by looking up the effect of the rare variants in our EUR-ancestry GWAS meta-analyses excluding UKBB participants, and
2. the gene-level by performing gene-burden testing in the deCODE and Genes & Health datasets.

Revised text:

Lines 520-521: “However, this association did not replicate in the deCODE or Genes & Health (G&H) datasets ($P > 0.05$, Supp. Tables 14, 20, and 21).”

Lines 525-542: “Across 43 gene-trait pairs with Cauchy $P < 5E-06$ in UK Biobank, thirteen (30.2%) replicate at nominal significance ($P < 0.05$), and two (4.65%) at Bonferroni-adjusted significance ($P < 6.85E-04$) in either the deCODE or G&H datasets with consistent directions of effect (Supp. Tables 14, 20, and 21). We replicated the testosterone-F lowering associations of rare damaging variation in the hydroxysteroid dehydrogenase enzymes *AKR1D1* (UKBB $P = 1.76E-08$, deCODE $P = 1.08E-07$, G&H $P = 0.862$) and *AKR1C3* (UKBB $P = 2.21E-09$, deCODE $P = 1.12E-06$, G&H $P = 8.75E-08$) in external cohorts ($P < 6.85E-04$, Bonferroni adjustment for 43 independent gene-trait pairs) (Supp. Tables 14, 20, and 21). We report the first known association of *HSD11B1* with testosterone-F (burden test $P = 1.93E-06$ when aggregated across missense variants with $MAF < 0.01\%$), with nominal replication in deCODE ($P = 0.028$); pathogenic variants in this gene are reported to cause hyperandrogenism due to cortisone reductase deficiency^{87,88} (Supp. Figure 11). We also report the association of testosterone-M with *HSD17B2* (burden test $P = 1.33E-11$ when aggregated across pLoF variants with $MAF < 0.1\%$), which encodes the enzyme hydroxysteroid 17 β -dehydrogenase 2 that catalyses the oxidation of oestradiol, testosterone, and dihydrotestosterone to less active forms and thus regulates the biological potency of steroid hormones^{89,90} (Supp. Figure 11 and Supp. Table 14). The association of rare damaging variation in *HSD17B2* with lower testosterone nominally replicated in deCODE ($P = 2.22E-03$) and G&H ($P = 0.0481$).”

Lines 553-562: “23 of these rare variants were present in our EUR-ancestry GWAS meta-analyses excluding UKBB participants, 21 (91.3%) of which had the same direction of effect across analyses (Supp. Table 15). Three rare variants associated with testosterone-F (in genes *ZKSCAN1*, *ZAN*, and *SHBG*), and one rare variant associated with testosterone-M (in *SHBG*) replicated in the GWAS meta-analysis at $P < 2.17E-03$ (Bonferroni adjustment for 23 variants tested); an additional six variants replicated at $P < 0.05$ (Supp. Table 15).”

Supp. Text (p6, para 1): “The testosterone-F association with *TNXB* replicated nominally in deCODE ($P = 0.041$); rare variants in *TNXB* cause Ehlers-Danlos syndrome¹⁶², a collection of disorders affecting connective tissues that has previously been linked to sex hormone dysregulation¹⁶³.”

Supp. Table 15: Now includes the GWAS meta-analysis look-ups of the rare variants.

Supp. Table 20: Replication of genes associated with infertility and reproductive hormones in deCode whole genome sequencing analyses.

Supp. Table 21: Replication of genes associated with infertility and reproductive hormones in Genes & Health whole exome sequencing analyses.

Minor comments:

The abbreviation of GWS for genome-wide signal seems unnecessary

We have replaced all seven instances of the abbreviation “GWS” with the full form: “genome-wide significant”.

Supp Table 1: it would be helpful to have text descriptions of what conditions/diseases were considered to fall into the different categories of infertility.

Thank you for this suggestion; we have substantially re-worked Supp. Table 1 to include text descriptions for all codes and display how the different categories of female infertility relate to each other.

Supp Table 2 and Supp Table 9 – please add to the paper the total sample size for the meta-analysis of each phenotype.

We have now included rows for total sample size of each phenotype in European and all-ancestry meta-analyses in Supp. Table 2 and Supp. Table 9.

Supp Table 3, Supp Table 10, Supp Table 15 – please add the effect allele and clarify which is the effect allele tested and the frequency of that allele. Likewise, Table 1 (i.e. is A1 the minor allele?).

Thank you for the suggestion to improve clarity on the effect allele. Table 1, and Supp. Tables 3, 10, and 15 have now all been modified to present the minor allele as effect allele (i.e. A1 is always the minor allele, which is the effect allele).

How were the genes mapped to the GWAS signals in Table 1?

We have now specified in the header for Table 1 (and the relevant Supp. Tables) that variants were mapped to their nearest genes as follows: exonic or intronic variants were assigned to their respective genes, and variants in intergenic regions were mapped to the nearest transcription start site (TSS).

Please add effect allele and frequency to discussion of variants in the text (e.g. to line 184-185).

We have added the minor allele (effect allele) frequency to all variants discussed in the text through **lines 190-222** and **lines 381-398**.

Did the genome-wide results show any evidence of genomic inflation? Useful to include LDSC intercept in Supp Table 6.

Thank you for this comment, we have now included the LDSC intercept and standard error in Supp. Table 6. In all cases, the LDSC intercept was close to 1 (the expected value when the study is unconfounded); any significant ($P < 0.05$) deviations from 1 were small. The largest inflation was observed for Testosterone-F (intercept (SE)=1.06 (0.0221), $P=0.00663$). The QQ-plots for Testosterone-F are attached below, showing reasonable control of confounding factors across the allele frequency spectrum.

Figure. Quantile-quantile plots for GWAS meta-analysis of Testosterone-F (EUR ancestry) within different minor allele frequency (MAF) bins. From top-left to bottom-right: MAF<0.01%, [0.01%-0.1%), [0.1%-1%), [1%-10%), and >=10%. The lambdaGC within each bin is also noted.

Line 307-310 – Why were other reproductive associations not considered?

Thank you for the comment. The specified line refers to how we classify hormone associations (novel for any reproductive hormone, novel for the specific hormone, or known). We were interested in the “novelty” of hormone SNPs in the context of hormone measurements, i.e. we think a hormone-associated SNP should be called “novel” even if it has previously been reported for a reproductive disease. This is because reproductive **diseases** are different from reproductive **hormones** in many important ways - for example, there is no reason to say a testosterone SNP is already known if it is associated with PCOS, as it may point to an important mechanism by which the SNP affects both testosterone and PCOS; but perhaps it should not be called “novel” if it is known to associate with SHBG, as the relationship between SHBG and testosterone levels is clear.

Line 349 – Figure 3 legend suggests that AMH is at P<0.05

Thank you for spotting this typo in the text, also discussed in response to Major Comment #2, which we have now edited on **line 439**.

Tables and Supplementary Tables: it would be helpful to state the Bonferroni corrected p-thresholds in the headers.

Thank you, we have added significance thresholds to Table 1 and Supp. Tables 3, 4, 5, 7, 8, 10, 11, 12, 13, 14, 15, 19, 20, 21, and 22.

Supp Table 13 it would be helpful to provide results for all 703 traits tested.

Thank you, all 702 phenotypes (excluding “is_female”, as discussed in response to Major Comment #4) are now included in Supp. Table 13.

Line 390 – I couldn't match the Cauchy F-EXCL p-value to Supp Table 14.

Thanks for spotting this typo, the correct P-value is 2.47E-07, which we've corrected (now on **line 518**).

Figure 6A – it was confusing that the points are sized by beta as well as having this on the y axis

Thank you for the recommendation. We have modified figure 6A to only show beta on the y-axis, with all points sized the same regardless of beta.

Supplementary Table 15 – it would be helpful to add the ID and MAF of the common variant

Thank you for this suggestion, we have added this information on the rsid and MAF of the common variant to Supp. Table 15.

Ref 116 – issue with formatting

We thank the reviewer for spotting this formatting error, which we have now edited (new **Ref 126**).

Supplementary Methods: This line does not make sense “GWASs in the UKBB were additionally adjusted for one-hot encoding of data provider or assessment centre”

We use “one-hot encoding” as the statistical term for transforming a categorical variable with K categories into K-1 binary variables; however, we appreciate this may be confusing language and have therefore deleted the phrase “one-hot encoding”.

Reviewer #2:**Remarks to the Author:**

I commend the authors for their novel, comprehensive, and very well-written study. I have no major concerns or issues with the presented research and provide the following comments in line with the *Nature Genetics* review process and goal to improve and/or extend the interpretation of the study findings.

Thank you for the kind feedback and for taking the time to generously review our work.

Summary of the key results

The key results are both appropriate and reasonable in their emphasis. The manuscript most certainly provides "the first comprehensive view of the genetic architecture of infertility".

However, I question the conclusion that their "results suggest that while individual genes associated with hormone regulation may be relevant for fertility, there is limited genetic evidence for correlation between reproductive hormones and infertility at the population level" (Page 3, Line 105-106). See 'Suggested improvements'.

Originality and significance: if not novel, please include reference

The research and findings are highly novel. In addition to the identified novel risk loci and genes, the GWAS summary statistics resulting from this study (made available through the GWAS Catalog upon publication) will prove extremely valuable for future research aimed at understanding the role and relationship of infertility and reproductive hormones in other human traits.

Data & methodology: validity of approach, quality of data, quality of presentation

The study utilised current and appropriate methodology, the data was of high quality and presented in a very clear and concise manner.

Appropriate use of statistics and treatment of uncertainties

The employed statistical tests were appropriate and correctly interpreted. The study was very well-powered and does not overstate its findings.

Conclusions: robustness, validity, reliability

The results are carefully interpreted and discussed with respect to current knowledge. The conclusion are appropriate, with perhaps the exception regarding "limited genetic evidence for correlation between reproductive hormones and infertility at the population level" (see 'Suggested improvements').

Suggested improvements: experiments, data for possible revision

Major comment:

The authors make numerous important statements regarding (absence of) correlation between traits at the population level that are based largely, if not solely, on LDSC genetic correlation (rg) analyses. However, given LDSC assesses the global (genome-wide) rg, it only considers and quantifies the average sharing of genetic effects between a pair of traits.

Therefore, when r_g is confined to a particular genomic region(s) or is in opposite directions at varying loci, LDSC may not detect a significant r_g .

Indeed, the authors state that “Only 9.80% (of 153 total) lead variants for testosterone in females and 5.75% (of 261 total) lead variants for testosterone in males reach GWS in both sexes; and 45.9% of variants have opposing directions of effect in men and women (Supp. Figure 6). Indeed, we found no significant genetic correlation between testosterone in men and women (r_g (SE)=0.0361 (0.0428), $P=0.399$).” (Page 13, Line 332-335). However, I would argue that 9.8% and 5.75% is a non-negligible (and significant) overlap of genome-wide significant risk loci. Furthermore, the observation that “45.9% of variants have opposing directions of effect in men and women” would perfectly explain why the r_g between testosterone in men and women was not significant (i.e., the averaging of positive and negative effects result in a non-significant r_g).

We thank the reviewer for this comment. We agree that the low r_G between testosterone in men and women may be explained by the averaging of positive and negative effects at some loci, as suggested by Supp. Figure 6A.

We do think it is unusual and worth highlighting that the genetic correlation between sexes for testosterone is null, and that nearly half of all loci exhibit significant differences in effect sizes between sexes. For most common complex traits, we would expect genome-wide significant loci to nearly completely overlap, with similar effect sizes between sexes (Bernabeu *et al.* (2021), who analysed sex differences among 161 traits in UK Biobank). We compared our results to Bernabeu *et al.* and report the below.

Revised text:

Lines 421-423: “... correlation between testosterone in men and women (r_g (SE)=0.0361 (0.0428), $P=0.399$), which is the lowest cross-sex genetic correlation among all traits in the UK Biobank (Supp. Text).”

Supp. Text (p3, para 2): “Bernabeu *et al.* (2021) calculated the genetic correlation between sexes for 161 traits in the UK Biobank¹⁶¹. Testosterone (cross-sex $r_G=3.61\%$) has the lowest cross-sex genetic correlation across all these traits; the next lowest are for self-reported ear/nose/throat disorders ($r_G=22.8\%$) and disorders of the urinary tract ($r_G=40.4\%$).”

Supp. Text (p3, para 3): “Bernabeu *et al.* (2021) also reported the number of sex-dimorphic lead SNPs (sdSNPs, defined as having sex-heterogeneity $P<1E-08$ for difference between male and female effect sizes) across 530 traits in the UK Biobank¹⁶¹. We followed the same protocol to define sdSNPs for testosterone. 104 traits (20%) assessed by Bernabeu *et al.*¹⁶¹ had at least one autosomal sdSNP; and only waist-to-hip ratio (100 sdSNPs) had more sdSNPs than testosterone (73 sdSNPs). The trait with the most sdSNPs after testosterone was height (18 sdSNPs) (Supp. Figure 16). Testosterone also had the most X-chromosome sdSNPs (6) of any trait evaluated in UK Biobank (Supp. Figure 16).”

Supp. Figure 16: Number of sex-dimorphic lead SNPs (sdSNPs) for testosterone as compared to other traits in UK Biobank.

Similarly, an over-reliance on LDSC r_g in determining a (polygenic) relationship between traits extends to the “limited genetic evidence for correlation between reproductive hormones and infertility” (Page 3, Line 105-106); “24 obesity-related traits, including body mass index (BMI), waist-to-hip ratio (WHR), and body fat percentage, are correlated with testosterone and FSH, but are not genetically correlated with any category of female infertility” (Page 14, Line 367-369); “we did not find substantial genetic correlation between obesity and infertility” (Page 17, Line 477-478); and “We did not find evidence that reproductive hormone dysregulation and obesity are strongly correlated with infertility at the population level, but instead nominate individual hormone-associated genes with effects on fertility” (Page 18, Line 525-527).

Thus, use of recently developed approaches such as LAVA (Local Analysis of [co]Variant Association) [Werme et al 2022 doi:10.1038/s41588-022-01017-y] could help examine local r_g . This will allow the examination of and provide insight into their complex genetics.

Also, the ‘MiXeR’ method quantifies polygenic overlap, using GWAS summary statistics, irrespective of (global) genetic correlation (Frei et al 2019 doi: 10.1038/s41467-019-10310-0).

Indeed, I suggest (argue) that in addition to “Other genetic and non-genetic avenues must be explored to treat complex and heterogeneous fertility disorders that impact the physical, emotional, and financial well-being of millions of individuals across the globe” (Page 19, Line 527-529) these additional genetic analyses (LAVA and MiXeR) should be performed for the current study.

We thank the reviewer for this valuable comment, based on which we implemented LAVA and MiXeR to examine local genetic correlations and polygenic overlap, respectively, between: female infertility (F-ALL, F-ANOV, F-INCL) and: 4 reproductive hormones (FSH, testosterone, TSH, AMH), 5 obesity-related traits (body fat percentage, BMI, waist circumference, hip circumference, and comparative body size at age 10 years), and 5 female reproductive conditions (endometriosis, PCOS, heavy menstrual bleeding, uterine fibroids, and dizygotic twinning).

The results of these analyses are summarised in Figure 4, Supp. Figure 17, and Supp. Tables 22-24. We have added the below to the text.

Revised text:

Lines 243-251: “We tested for local bivariate genetic correlations between infertility and PCOS, endometriosis, heavy menstrual bleeding, and uterine fibroids at 2,495 blocks across the genome, chosen to be approximately 1Mb in length each while minimising LD between blocks⁵¹. Consistent with the genome-wide r_g , we found positive local r_g between female infertility and reproductive disorders at 11 regions ($P < 1.91E-05$, Bonferroni adjustment for 2,618 local bivariate tests performed at regions with significant heritability of both traits in each pair tested; Figure 4A and Supp. Table 22). At 5/11 blocks, infertility was correlated with more than one reproductive condition, none of which had individual effects after conditioning upon the other associated reproductive disorders in the region (all $P > 0.05$, Supp. Table 22).”

Lines 253-265: “Furthermore, we used MiXeR⁵² to assess bivariate polygenic overlap, regardless of genome-wide genetic correlation, between infertility and reproductive conditions (Supp. Text). We found that approximately 50% of causal SNPs involved in endometriosis and the assessed infertility phenotypes were shared, with varying degrees of effect-size correlation in the shared polygenic component (F-INCL: 563 shared causal SNPs of 1,053 total, ρ (SE)=0.957 (0.0649) in the shared polygenic component; F-ALL: 721/1,448, ρ (SE)=0.544 (0.298); F-ANOV: 562/1,150, ρ (SE)=0.141 (0.0793)) (Figure 4B, Supp. Table 24). Similarly, we found that between 23.7%-26.4% of causal SNPs involved in infertility and uterine fibroids were shared, with genetic correlation in the shared component ranging from ρ =0.431 (SE=0.183) with F-ANOV to ρ =0.738 (0.112) with F-ALL. We noted that while there was substantial correlation in the shared component of F-ANOV and PCOS (ρ (SE)=0.878 (0.242)), only 97 (10.9%) of the 888 causal variants involved were shared; the majority (88.2%) of variants were unique to F-ANOV and only 8 variants (<1%) were unique to PCOS (Figure 4B, Supp. Table 24).”

Lines 267-280: “We observed genome-wide negative correlation between F-ANOV and spontaneous dizygotic twinning, a heritable metric of female fecundity that captures the propensity for multiple ovulation⁵⁰ (r_g =-0.740 (0.182), P =4.93E-05). We also found substantial negative correlation in the shared polygenic component of these traits (ρ (SE)=-0.920 (0.129)), with 32% (295) shared SNPs of the 912 total causal SNPs involved (Figure 4B, Supp. Table 24); however, there were no significant local genetic correlations between infertility and dizygotic twinning after Bonferroni adjustment for multiple testing (all P >1.91E-05). Despite the lack of genome-wide correlation between dizygotic twinning and other definitions of infertility, and limited polygenic overlap (13.3%-19.1%), we observed strong positive correlation between twinning and F-ALL (ρ =0.752 (0.310)) and F-INCL (ρ =0.921 (0.146)) in their shared polygenic components and local correlation in the chr22:44409534-45170649 region (F-ALL local r_g = 0.632 (0.163), P =2.43E-04) (Figure 4B, Supp. Table 24). These local effects may arise from the residual influence of age at some loci – although fertility in women declines with age, the propensity for double ovulation also increases with age⁵³.”

Lines 446-457: “The limited genetic correlation between infertility and reproductive hormones was mirrored in polygenic overlap analyses. The highest proportion of shared SNPs between these traits was 14.5% between F-ANOV and testosterone (209/1,444 variants shared, ρ (SE)=0.549 (0.252) in the shared polygenic component), followed by 14.0% between F-ANOV and AMH (123/881, ρ (SE)=0.993 (0.0301)) (Figure 4B, Supp. Table 24). That said, nearly all (123/124) causal variants for AMH were also in the polygenic component for F-ANOV, as were a substantial proportion of causal SNPs for FSH (90/99), which were both traits with low polygenicity (Supp. Text). Both F-ALL and F-INCL shared fewer than 10% of putative causal variants with any reproductive hormone (Supp. Table 24). However, despite minimal overlap (<6.69%, or 111 of 1,666 SNPs involved), there was substantial negative correlation in the shared genetic components between TSH and F-ALL (ρ (SE)=-0.806 (0.149)) and F-INCL (ρ (SE)=-0.659 (0.314)) (Figure 4B, Supp. Table 24).”

Lines 459-480: “Consistent with the genome-wide results, we also found no evidence for local genetic correlations between any category of infertility and the above hormones across the genome at Bonferroni-adjusted significance threshold (all P >1.91E-05, Figure 4A and

Supp. Table 22). At a relaxed significance threshold of FDR-adjusted $P < 0.05$, 14 genomic regions contain local bivariate genetic correlations, including negative associations between testosterone and F-ALL (local r_g (SE) = -0.473 (0.140)), F-ANOV (r_g = -0.490 (0.114)), and F-INCL (r_g = -0.423 (0.117)) at the chr17:7264459-8554763 locus which encompasses 64 genes including *SHBG* (Supp. Text)."

Lines 482-496: "Consistent with the genome-wide results, we found no evidence for local genetic correlations between any category of infertility and five obesity-related traits at 2,495 regions across the genome at a Bonferroni-adjusted significance threshold (all $P > 1.91E-05$, Figure 4A and Supp. Table 22). However, at a relaxed significance threshold (FDR-adjusted $P < 0.05$), we identified 18 genomic regions with local genetic correlations between infertility and obesity, including three regions where multiple obesity-associated traits were independently associated with infertility (Supp. Text). Polygenic analyses also revealed only limited overlap between infertility and obesity: fewer than 10% of causal SNPs involved were shared between infertility and any of the five obesity-related traits assessed (Figure 4B, Supp. Table 24, Supp. Text). The low overlap may reflect the polygenicity of obesity (between 4,050-11,000 causal variants), of which the majority (between 73.2%-93.0%) are not involved in infertility (Supp. Tables 23-24). However, despite this minimal overlap, there was substantial negative correlation in the shared genetic components between F-INCL and comparative body size at age 10 years (451 shared SNPs of 4,385 involved, ρ (SE) = -0.874 (0.143)), and adult BMI (393/11,185, ρ (SE) = -0.640 (0.262)) (Figure 4B, Supp. Table 24)."

Supp. Text (p4-p6): two sections titled: "Local genetic correlations between infertility, reproductive conditions, hormones, and obesity" and "Polygenic overlap between infertility, reproductive conditions, hormones, and obesity" to present and interpret: (1) FDR-significant results that do not reach Bonferroni significance, (2) univariate MiXeR results, and (3) detailed breakdown of bivariate results that are summarised in the Main text.

Figure 4: Local genetic correlations and polygenic overlap between female infertility and other phenotypes.

Supp. Figure 17: Number of blocks with FDR-significant bivariate local genetic correlations between female infertility (y-axis) and reproductive diseases, reproductive hormones, or obesity (x-axis).

Supp. Table 22: Local genetic correlations between female infertility and reproductive hormones, reproductive conditions, and obesity, as estimated using LAVA.

Supp. Table 23: Univariate MiXeR results.

Supp. Table 24: Bivariate MiXeR results.

Minor comments:

It was not clear how the lead "SNPs were annotated to the mapped gene" - can the authors please add these details.

We have now specified in the header for Table 1 (and the relevant Supp. Tables) that variants were mapped to their nearest genes as follows: exonic or intronic variants were assigned to their respective genes, and variants in intergenic regions were mapped to the nearest transcription start site (TSS).

Re: the statement “Finally, we identified 29 unique genes carrying rare variants (MAF<1%) associated with testosterone in male or female participants in the UK Biobank. Eighteen of the 29 genes also contain common testosterone-associated variants from GWASs (MAF>1%), but the rare variant has a larger absolute effect size in the majority (83%) of these” (Page 15, Line 413-416): could the authors please comment on whether the common and rare variants at the same loci are statistical independent and/or are perhaps the result of synthetic associations (e.g., see Wray et al 2011 doi: 10.1371/journal.pbio.1000579)?

We thank the reviewer for this note. We realised during review that there was an error in identifying lead variants on the X chromosome, which increases the number of loci with nearby common variants to 19 (from 18). We conditioned the rare-variant associations on nearby lead common variants from GWASs (within 500 kb of the rare variant with MAF>1%) and report that 14/19 rare variants retain their effect ($P<1E-07$) even after conditioning on common variants. We have added a column to Supp. Table 15 with effect sizes of rare variants after conditioning on nearby common variants, and the following text.

Revised text:

Lines 559-562: “Nineteen of the 29 genes also contain nearby (+/- 500kb) common testosterone-associated variants from GWASs (MAF>1%), but at the majority (74%) of these loci, the effect of the rare variant is larger and remains upon conditioning on common variants ($P<1E-07$ after conditioning, Figure 6A, Supp. Table 15, and Supp. Text).”

Supp. Text (p6, para 2): “Three rare variants on chr7 associated with testosterone-F (in genes *ZKSCAN1*, *GPC2*, and *STAG3*) are no longer significant (all $P>1E-07$) after conditioning upon nearby common variant rs17250196 (MAF=5.18%, $\beta=-0.129$, $P=4.90E-84$). Similarly, five rare variants on chr17 associated with testosterone-M (in genes *SLC2A4*, *NLGN2*, *SPEM1*, *SPEM2*, and *TMEM102*) do not remain significant after conditioning upon rs12946520 (MAF=36.3%, $\beta=0.113$, $P=0$); and the effect of a rare variant in the androgen receptor gene on chrX may similarly be explained by two nearby common variants, rs139478468 (MAF=7.89%, $\beta=0.0588$, $P=3.26E-55$) and rs7052964 (MAF=18.8%, $\beta=0.0403$, $P=1.65E-53$).”

Supp. Table 15: now includes a column with effect sizes of rare variants following conditioning on nearby common variants.

The statement "The strong genetic correlation of 71% between idiopathic infertility and endometriosis may indicate that some proportion of idiopathic cases are due to under-diagnosis of endometriosis, whose early treatment may prevent future infertility" (line 467-469)

The reviewer has withdrawn this comment.

References: appropriate credit to previous work?

The references appear appropriate and sufficient.

Clarity and context: lucidity of abstract/summary, appropriateness of abstract, introduction and conclusions

All aspects of the manuscript are very clear and contextualised.

Reviewer #1:**Remarks to the Author:**

This remains an interesting and important paper that will make a valuable addition to the field. I would like to thank the authors for their thorough and clear responses and extensive additional analyses.

One minor comment:

Line 424: I suggest re-wording slightly – as I understand it, in the UK FSH is still used to diagnose premature menopause but is not recommended for diagnosis of menopause in the normal age range. See <https://cks.nice.org.uk/topics/menopause/diagnosis/diagnosis-of-menopause-perimenopause/>

Otherwise, I have no further comments.

We thank the reviewer for this suggestion. Indeed, FSH is still used to diagnose premature menopause, so we have revised the text (now on line 289) to read: “Clinical measurements of FSH and LH may be used to diagnose premature menopause,...”

Reviewer #2:**Remarks to the Author:**

I thank the authors for their careful and detailed responses to my comments. The authors have performed additional analyses (as per my suggestions/requests) and appropriately amended their manuscript to my satisfaction. The resulting revised manuscript is very balanced and scientifically robust. I have no further comments, other than to say: "well done, nice work!".

We thank the reviewer for their kind words.